# Vacancies tailoring lattice anharmonicity of Zintl-type thermoelectrics

Jinfeng Zhu[1,18], Qingyong Ren[2,3,4,18] ✉, Chen Chen[5,6,18], Chen Wang[7], Mingfang Shu[1], Miao He[8,9], Cuiping Zhang[1], Manh Duc Le[10], Shuki Torri[11], Chin-Wei Wang[12], Jianli Wang[13,14], Zhenxiang Cheng[14], Lisi Li[2,3,4], Guohua Wang[1], Yuxuan Jiang[15], Mingzai Wu[15], Zhe Qu[8,9], Xin Tong[2,3,4] ✉, Yue Chen[7] ✉, Qian Zhang[5,16] ✉ & Jie Ma[1,17] ✉

While phonon anharmonicity affects lattice thermal conductivity intrinsically and is difficult to be modified, controllable lattice defects routinely function only by scattering phonons extrinsically. Here, through a comprehensive study of crystal structure and lattice dynamics of Zintl-type Sr(Cu,Ag,Zn)Sb thermoelectric compounds using neutron scattering techniques and theoretical simulations, we show that the role of vacancies in suppressing lattice thermal conductivity could extend beyond defect scattering. The vacancies in $Sr_2ZnSb_2$ significantly enhance lattice anharmonicity, causing a giant softening and broadening of the entire phonon spectrum and, together with defect scattering, leading to a ~ 86% decrease in the maximum lattice thermal conductivity compared to SrCuSb. We show that this huge lattice change arises from charge density reconstruction, which undermines both interlayer and intralayer atomic bonding strength in the hierarchical structure. These microscopic insights demonstrate a promise of artificially tailoring phonon anharmonicity through lattice defect engineering to manipulate lattice thermal conductivity in the design of energy conversion materials.

Thermal conductivity is a fundamental physical property of condensed matter and extreme thermal conductivity is highly desirable for many applications[1-4]. In particular, ultralow lattice thermal conductivity facilitates the design of energy conversion or harvesting materials, such as sustaining large temperature gradients in thermoelectric converters[5] or generating strong thermal localization in optoacoustic perovskites[6]. In these insulator- or semiconductor-based scenarios, the primary heat carriers are the phonons, a type of quantized

[1]Key Laboratory of Artificial Structures and Quantum Control, School of Physics and Astronomy, Shanghai Jiao Tong University, Shanghai, China. [2]Institute of High Energy Physics, Chinese Academy of Sciences, Beijing, China. [3]Spallation Neutron Source Science Center, Dongguan, China. [4]Guangdong Provincial Key Laboratory of Extreme Conditions, Dongguan, China. [5]School of Materials Science and Engineering, Harbin Institute of Technology, Shenzhen, China. [6]School of Physical Sciences, Great Bay University, Dongguan, Guangdong, China. [7]Department of Mechanical Engineering, The University of Hong Kong, Hong Kong SAR, China. [8]Anhui Province Key Laboratory of Low-Energy Quantum Materials and Devices, CAS Key Laboratory of Photovoltaic and Energy Conservation Materials, High Magnetic Field Laboratory of Chinese Academy of Sciences (CHMFL), HFIPS, CAS, Hefei, China. [9]Science Island Branch of Graduate School, University of Science and Technology of China, Hefei, China. [10]ISIS Neutron and Muon Source, Rutherford Appleton Laboratory, Chilton, Didcot, Oxon, England, UK. [11]Institute of Materials Structure Science, High Energy Accelerator Research Organization (KEK), Tokai, Ibaraki, Japan. [12]Neutron Group, National Synchrotron Radiation Research Center, Hsinchu, Taiwan. [13]College of Physics, Jilin University, Changchun, China. [14]Institute for Superconducting and Electronic Materials, Faculty of Engineering and Information Sciences, University of Wollongong, Innovation Campus, North Wollongong, Australia. [15]School of Physics and Optoelectronics Engineering, Anhui University, Hefei, Anhui, China. [16]State Key Laboratory of Advanced Welding and Joining, Harbin Institute of Technology, Harbin, China. [17]Collaborative Innovation Center of Advanced Microstructures, Nanjing 210093 Jiangsu, China. [18]These authors contributed equally: Jinfeng Zhu, Qingyong Ren, Chen Chen. ✉e-mail: renqy@ihep.ac.cn; tongxin@ihep.ac.cn; yuechen@hku.hk; zhangqf@hit.edu.cn; jma3@sjtu.edu.cn

quasiparticles of the normal modes of lattice vibration. Managing lattice thermal conductivity by engineering phonons is becoming increasingly important in the fields of condensed matter, materials science, and electronic engineering[7].

Within the classical Debye-Callaway model of free phonon gases, the phonon-dominated lattice thermal conductivity can be simply written as $\kappa_{lat} = 1/3c_V v^2 \tau$, where $c_V$ is the lattice heat capacity, $v$ is the phonon group velocity and $\tau$ is the phonon relaxation time or lifetime. One typical strategy to slow down heat transport is to develop materials with intrinsically low lattice thermal conductivity[8]. Complex crystal structure, heavy atomic mass, and low bonding strength cause small heat capacity and group velocity of acoustic phonons, which are the main heat-conducting channels[9]. Furthermore, stereochemically active lone pair of $ns^2$ electrons[10], weak chemical bonding[11,12], resonant bonding[13], rattler atomic vibrations[14,15], and four-phonon Fermi resonance[16] always induce strong phonon anharmonicity and introduce Umklapp scattering, which can significantly shorten the phonon lifetime[8]. All of the above factors are intrinsic properties and mainly depend on a given material's crystal symmetry and chemical components. On the other hand, importing extrinsic defects is a controllable strategy to scatter phonons and suppress lattice thermal conductivity[17], for example, zero-dimension (0D) point defects, 1D linear dislocations, 2D planar interfaces, and 3D body defects[9,18–20]. The phonon scattering strength of those crystal imperfections strongly depends on their density in the crystal matrix, hence, leaves the possibility to artificially engineer phonons and tune the lattice thermal conductivity[17,18,21].

The 0D point defects can be achieved in the form of heteroatom substitution, interstitial filling, high-entropy alloying, or vacancies in thermoelectric materials[17,22]. Most of them are traditionally regarded as important phonon scattering mechanisms in thermal transport, originating from mass and strain fluctuations, while their influences on phonon energy and group velocity have been overlooked[23–27]. Nonetheless, recent studies of the SnTe-based compounds implied that vacancies might be responsible for lattice softening[28,29]. More interestingly, several half-Heusler compounds (e.g. $Nb_{0.8}CoSb$, $Ti_{0.9}NiSb$, and $V_{0.9}CoSb$) and Zintl phases (e.g. $Sr_2ZnSb_2$ and $Eu_2ZnSb_2$) are recently found to contain high concentrations of vacancies and exhibit much lower lattice thermal conductivity than their counterparts without vacancies[30–33]. The large number of vacancies in these compounds does not create long-range ordering, which would form a superlattice and then induce a change in crystal symmetry, as is the case in the well-known $Cs_2SnI_6$-based double perovskites[34–36]. Rather, these vacancies may exhibit intriguing short-range ordering, local ordering, or random distribution[37,38]. Although the impact of the vacancies in these unique materials on suppressing lattice thermal conductivity is believed to go beyond as defect scattering of phonons based on bulk properties, a complete atomic-level understanding is still missing.

Herein, we report a comprehensive study of the effects of vacancies on the lattice thermal conductivity of ZrBeSi-type Zintl system as SrCuSb, SrAgSb, and $Sr_2ZnSb_2$. The crystallographic structures are measured with neutron powder diffraction (NPD), the lattice dynamics are mapped and studied using the inelastic neutron scattering (INS) technique and molecular dynamic simulations, and the variations of atomic bonding caused by vacancies are analyzed with X-ray photoelectron spectroscopy (XPS), and theoretical calculations. We reveal that the large number of vacancies in the $Sr_2ZnSb_2$ compound leads to an overall softening of the entire crystal lattice and a larger phonon anharmonicity in comparison with SrCuSb and SrAgSb. Furthermore, this dramatic change in the lattice dynamics arises from the weakening ionic bonding between the Sr and [ZnSb] sublattices. These findings demonstrate that the vacancies in the Zintl-phase compounds play more complicated roles beyond phonon-defect scattering in the suppression of lattice thermal conductivity.

## Results

### Crystallographic structures and thermal transport properties

The Sr(Cu,Ag,Zn)Sb compounds crystallize in a hexagonal structure ($P6_3/mmc$) and have a honeycomb layered configuration with alternating Sr sublattice and [(Cu,Ag,Zn)Sb] sublattice repeating along the c-axis as shown in Fig. 1a–f. The NPD patterns at 300 K in Fig. 1g can be indexed by this hexagonal structure without discernible impurities. Rietveld refinements of the NPD patterns reveal that the 2d-site atomic position in the Sr(Cu,Ag)Sb compounds is fully occupied by Cu (or Ag), while only half is filled by Zn with 50% vacancies for $SrZn_{0.5}Sb$ (denoted as $Sr_2ZnSb_2$ hereafter, see more details in Supplementary Fig. 2 and Supplementary Table 1). The vacancies are introduced to maintain charge balance as Cu/Ag is replaced by aliovalent Zn, similar to the Eu counterpart[32,38]. In addition, a random distribution model of the vacancy on the 2d-site could perfectly fit the $Sr_2ZnSb_2$ NPD patterns, including all the peak positions and peak intensities, as shown in Supplementary Fig. 2c and Fig. 6, indicating an absence of long-range ordering of the vacancies. This differs from the well-known $Cs_2SnI_6$ double perovskite where vacancies take a long-range ordering and lead to a change in crystal symmetry and an emergence of superlattice Bragg peaks[35,36]. In the following context, it will show that the lattice dynamical simulation with random vacancy distribution could give a good reproduction of the experimental phonon density of states (DOSs).

The substitution of Cu by Ag or Zn induces great suppression on the lattice thermal conductivity. Specifically, the thermal transport measurements over 10 K to 280 K in Fig. 1h demonstrate that heavier Ag causes the lattice thermal conductivity decrease ~51% from the maximum value of ~6.82 W m⁻¹ K⁻¹ in SrCuSb to ~3.31 W m⁻¹ K⁻¹ in SrAgSb at ~50 K. Nonetheless, the introduction of a large number of vacancies by aliovalent substitution of Cu with Zn yields a much larger suppression by ~86% (from ~6.82 W m⁻¹ K⁻¹ to ~0.98 W m⁻¹ K⁻¹ at ~50 K). The lattice thermal conductivity even reaches as low as ~0.63 W m⁻¹ K⁻¹ at 280 K. This trend is consistent with the results above 300 K measured using the laser flash method[39]. Although the vacancies have been invoked to explain this ultralow lattice thermal conductivity[32,38,39], a detailed atomistic understanding still remains elusive due to the lack of lattice dynamic measurements and in-depth analysis of this ZrBeSi-type Zintl thermoelectric materials.

### Vacancy inducing anomalies in lattice and thermal properties

To explore the underlying mechanism of the vacancies on suppressing the lattice thermal conductivity in the Sr(Cu,Ag,Zn)Sb Zintl phases, we performed temperature-variable NPD and thermal property measurements. Rietveld refinements of the NPD data are displayed in Supplementary Figs. 4-6. All three compounds exhibit thermal expansion of the lattice-constants and the unit cell volume, as shown in Supplementary Fig. 7a–f and Supplementary Table 3. On the other hand, isothermal comparisons found that the unit cell of both SrAgSb and $Sr_2ZnSb_2$ expands in the ab-plane but contracts along the c-axis referring to the Cu compound (see for example the 300 K values marked in Fig. 1a–f). Interestingly, the unit cell volumes of both compounds are larger than that of SrCuSb. It is simple to understand the volume expansion in SrAgSb as the Ag has a larger covalent atomic radius (1.53 Å) than Cu (1.38 Å). However, a similar attempt failed in the $Sr_2ZnSb_2$ case of smaller covalent radius (Zn ~ 1.31 Å). Furthermore, the three samples were found to exhibit different thermal expansion rates (Supplementary Fig. 7g–i). The a-axis expands slower than the c-axis in SrCuSb, both axes almost proportionally expand in SrAgSb, while the a-axis expands faster in $Sr_2ZnSb_2$ (Supplementary Fig. 7i).

We further analyzed the overall isotropic atomic displacement parameters (ADPs), $B_{ov}$, to perceive the differences in atomic bonding and lattice thermal vibrations in Sr(Cu,Ag,Zn)Sb. Fitting of $B_{ov}$ to the modified Debye−Einstein model (Supplementary Note 1 and Fig. 2a−c) generates a Debye temperature, $\Theta_D$, of 279, 248, and 218 K for the Cu-, Ag-, and Zn-compounds, respectively, indicating the lattices for both

Ag- and Zn-compounds exhibit obvious softening with respect to Cu-compound. This agrees well with the $\Theta_D$ values extracted from the following analyses of the experimental sound velocities and heat capacities ($C_P$) (Table 1 and Fig. 2d–f). However, the absolute value of $B_{ov}$ in the Ag-compound is close to that in the Cu-compound, while the value in the Zn-compound is much larger. This suggests that the origins of the lattice softening in SrAgSb and $Sr_2ZnSb_2$ are different.

For polyatomic compounds, the contribution of optical phonons to the thermal properties should be included[40,41]. This necessity is demonstrated through the appearance of a Boson-like hump in the $C_P/T^3$ vs $T$ plot in Fig. 2g, since this hump feature implies a deviation of $C_P$ from the Debye $T^3$ law. Furthermore, the hump peak position in the $C_P/T^3$ vs $T$ plot corresponds to a crossover of the contribution to heat capacity from the acoustic phonons to optical phonons, and the lower peak positions in Fig. 2g indicate lower energies of the optical phonons in both the Ag and Zn compounds than in the Cu compound. The $\Theta_D$ and $\Theta_E$ (Einstein temperatures) from the fit of the $B_{ov}$ and the $C_P$ data in Fig. 2 are listed in Table 1. A careful comparison of the three Einstein temperatures, $\Theta_{Ei}^{HC}$, derived from $C_P$ (Supplementary Note 2) found a non-trivial discrepancy. Only the $\Theta_{E2}^{HC}$ value from the middle Boson peak shows an obvious drop in SrAgSb (in comparison with SrCuSb), while all the three $\Theta_{Ei}^{HC}$ values in $Sr_2ZnSb_2$ present large decrease. All these anomalies in the crystal structure and thermal properties imply that the introduction of vacancies induce different changes in atomic bonding and lattice dynamics from that of the introduction of Ag atoms.

## Lattice dynamics and strong phonon anharmonicity
To gain deep insights into the lattice dynamics of the Sr(Cu,Ag,Zn)Sb compounds, we performed INS measurements on the MARI time-of-flight chopper spectrometer at the ISIS Neutron and Muon Source, UK. The dynamic structure factor, $S(Q,E)$, as a function of momentum transfer $Q$ and energy transfer $E$, at 300 K are shown in Fig. 3a-c. The low-lying flat phonon band in SrCuSb, as marked with the pink dashed lines in Fig. 3a, mainly comprises the top of acoustic phonons away from the zone center and low-energy optical branches (Supplementary Fig. 9) and lies in the range of ~6.4 to ~9.5 meV, leaving a clear gap from its bottom energy to the elastic line (yellow arrow). However, this flat band moves to lower energy in SrAgSb and $Sr_2ZnSb_2$ and its bottom energy reduces to ~3.5 meV in both samples. This change becomes clearer in the neutron-weighted phonon density of states (DOSs), highlighted using the yellow shading in Fig. 3d,e, confirming that both SrAgSb and $Sr_2ZnSb_2$ have strong phonon softening relative to SrCuSb. In addition, it is noted that the phonon DOSs from 0 to ~3.5 meV also show obvious softening in energy or large increase in intensity, demonstrating a lowering of the acoustic phonon group velocities and agreeing with the experimental results of sound velocities and the Debye temperatures in Table 1.

Nevertheless, it is easy to spot differences in the lattice dynamics of SrAgSb and $Sr_2ZnSb_2$. Three sharp phonon bands can be clearly traced in the Ag compound, and the shift of the low-lying phonon band towards lower energy creates a clearer gap from the middle phonon band (centering at ~13 meV). However, all phonon bands become quite broader in the Zn compound, and it is not easy anymore to distinguish the phonon bands from each other. Moreover, the sound velocity analysis yields close Grüneisen parameters for SrCuSb (1.42) and SrAgSb (1.43), but the $Sr_2ZnSb_2$ gives a larger value of 1.50 (Table 1 and Supplementary Note 3). This enlarged behavior of the phonon

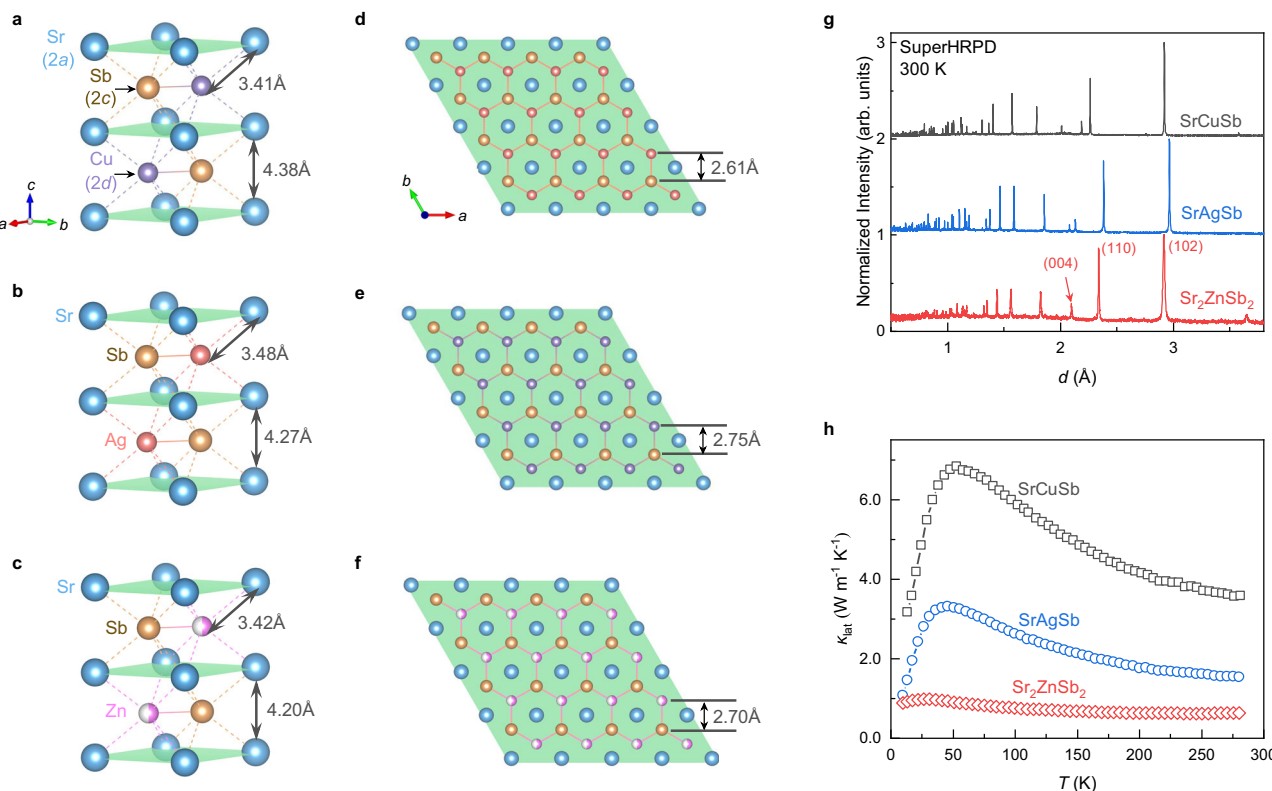

**Fig. 1 | Crystallographic structures and thermal transport properties of Sr(Cu,Ag,Zn)Sb.** a–c The hexagonal crystal structures of Sr(Cu,Ag,Zn)Sb with the same space group of $P6_3/mmc$, determined from the Rietveld refinement of NPD patterns collected on the Super High-Resolution Powder Diffractometer, SuperHRPD, at 300 K. The Wyckoff position notations are given after the atomic names in **a**. The 2d positions are partially and randomly occupied by Zn in $Sr_2ZnSb_2$. **d–f** A projected view of the crystal structures in the ab-plane. **g** NPD patterns at 300 K. The patterns are presented in the plane distance, d, space. The (102), (110) and (004) Bragg peaks are labeled. **h** Lattice thermal conductivity, $\kappa_{lat}$, over the temperature range of 10 to 280 K. Electrical transport properties, total thermal conductivities, and electrical thermal conductivities are shown in Supplementary Fig. 3. Source data are provided as a Source Data file.

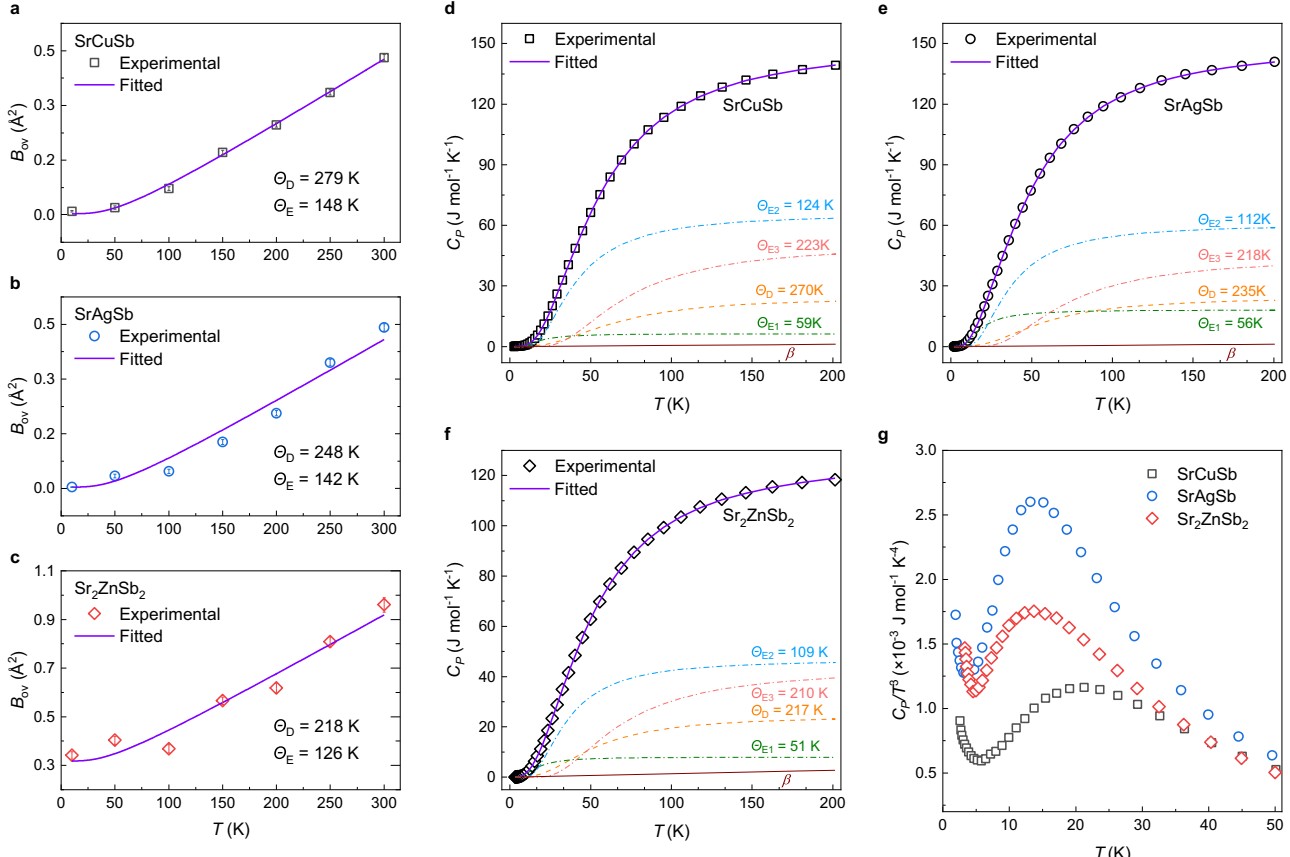

**Fig. 2 | Anomalies in atomic displacement parameters and heat capacities.**
**a–c** Overall isotropic atomic displacements, $B_{ov}$, of the Sr(Cu,Ag,Zn)Sb compounds fitted with a modified Debye−Einstein model (Supplementary Note 1 and Supplementary Tables 4). The error bars are estimated from the standard deviation of the Rietveld refinements. **d–f** Heat capacities, $C_P$, of Sr(Cu,Ag,Zn)Sb fitted with one Debye mode, three Einstein modes plus an individual contribution from electrons (Supplementary Note 2 and Supplementary Table 5). The Debye and three Einstein

modes are denoted as D, E1, E2 and E3, respectively. The Debye and Einstein temperatures, $\Theta_D$ and $\Theta_E$, are given in (**a–f**) from the fittings of $B_{ov}$ and $C_P$. $\beta$ represents the electronic contribution to the heat capacity. **g** $C_P/T^3$ vs $T$ (2–50 K) plot shows broad peak, demonstrating the necessity of the combined Debye-Einstein model in the fitting (Supplementary Fig. 8). Source data are provided as a Source Data file.

Grüneisen parameter in $Sr_2ZnSb_2$ is also captured by the average Grüneisen parameters, $\bar{\gamma} = d\ln\langle E\rangle/d\ln V$, where $\langle E\rangle = \int Eg(E)dE$ is the average phonon energy (Fig. 3f) and V is unit cell volume determined from NPD patterns. Here, the different absolute values of the Grüneisen parameter from these two methods might come from the oversimplified Debye model used in the sound velocity method. Nonetheless, these differences in lattice dynamics further demonstrate that the vacancies in $Sr_2ZnSb_2$ exert different influences on the lattice.

These features observed in the experimental data could be perfectly captured by our theoretical simulations based on the first-principles density function of theory (DFT) or equilibrium molecular dynamics (EMD) calculations (Fig. 3e). To rationally assign the measured phonon modes, the partial phonon DOSs is further analyzed in Fig. 3g–i. In comparison to SrCuSb, only the Ag-related phonon modes below ~10 meV exhibit obviously softening in SrAgSb while the energy for other modes does not make large variations. In contrast, all acoustic and middle-energy optical phonon modes in $Sr_2ZnSb_2$ show obvious softening and almost all phonon modes present large broadening, successfully reproducing the experimental results in Fig. 3c,d. Obviously, the lattice dynamics in the vacancy defective $Sr_2ZnSb_2$ differ from that in SrAgSb and SrCuSb.

**Weakening atomic bonding**
To rationalize the abnormal lattice dynamical behaviors of the vacancy defective $Sr_2ZnSb_2$, we examined the atomic bonds of $Sr_2ZnSb_2$ and

Sr(Cu,Ag)Sb through charge density difference and Bader charge analyses based on DFT calculations[42]. Figure 4a–i visualize the results of a comparative study of the charge density differences in Sr(Cu,Ag,Zn)Sb. It is obvious that the Cu(Ag) and Sb atoms in the [Cu(Ag)Sb] sublattice of SrCuSb (or SrAgSb) are stitched together through strong bonding as enhanced density is seen at the midpoint of each Cu(Ag)-Sb pair. On the other hand, the clearly separated charge distribution between the [Cu(Ag)Sb] sublattice and Sr sublattice indicates an interlayer ionic bonding nature. Furthermore, within each [Cu(Ag)Sb] layer, there exist polyatomic-central-shared charges itinerantly surrounding the [Cu(Ag)Sb] honeycomb (Fig. 4d,e). These broadly shared charges manifest as a peanut in the (110)-plane residing around the Cu(Ag) atoms (Fig. 4a,b). These features confirm that SrCuSb and SrAgSb follow the Zintl-phase concept[43].

However, the replacement of Cu or Ag by Zn with one more valence electron leaves half of the Zn crystallographic position as vacancies in order to maintain valence balance. These changes subtly regulate the atomic bonds. Firstly, the locally shared charges within Cu(Ag)-Sb pair in Sr(Cu,Ag)Sb move from Zn to Sb in $Sr_2ZnSb_2$ (see Fig. 4a-f). This could be explained by a large electronegativity difference between the Sb (2.05) and Zn (1.65) atoms, while the electronegativity of Sb is quite close to that of Cu (1.90) or Ag (1.93). Therefore, the atomic bond within the [(Cu,Ag,Zn)Sb] sublattice transfers to a more ionic nature following the introduction of Zn and vacancies. In addition, the vacancies could break down the bond in the

**Table 1 | Physical properties of SrCuSb, SrAgSb and Sr$_2$ZnSb$_2$, including sound velocities, Grüneisen parameters, Debye and Einstein temperatures**

| Physical properties | SrCuSb | SrAgSb | Sr$_2$ZnSb$_2$ |
|---|---|---|---|
| $v_l$ (m s$^{-1}$) | 4237 | 3803 | 3767 |
| $v_s$ (m s$^{-1}$) | 2500 | 2236 | 2174 |
| $\gamma_a$ | 1.42 | 1.43 | 1.50 |
| $\bar{\gamma}$ | 0.769 | 1.739 | 2.799 |
| **Debye Temperatures** | **SrCuSb** | **SrAgSb** | **Sr$_2$ZnSb$_2$** |
| $\Theta_D^{ADP}$ (K) | 279(7) | 248(7) | 218(8) |
| $\Theta_D^{sv}$ (K) | 279 | 243 | 227 |
| $\Theta_D^{HC}$ (K) | 270(3) | 235(4) | 217(2) |
| **Einstein Temperatures** | **SrCuSb** | **SrAgSb** | **Sr$_2$ZnSb$_2$** |
| $\Theta_E^{ADP}$ (K) | 148(1) | 142(3) | 126(7) |
| $\Theta_E^{HC}$ (K) | 135(4) | 129(3) | 123(7) |
| $\Theta_{E1}^{HC}$ (K) | 59(5) | 56(2) | 51(5) |
| $\Theta_{E2}^{HC}$ (K) | 124(4) | 112(4) | 109(5) |
| $\Theta_{E3}^{HC}$ (K) | 223(4) | 218(5) | 210(5) |

The Debye and Einstein temperatures were derived from the analyses of sound velocity (SV), overall isotropic ADPs, and heat capacities (HC) measurements (Supplementary Notes 1-3). The longitudinal and shear sound velocities, $v_l$ and $v_s$ were measured at room temperature, yielding Grüneisen parameters, $\gamma_a$. The average Grüneisen parameters, $\bar{\gamma}$, are also estimated from the experimental phonon DOSs data. More detailed results from the analyses of ADPs and HC are summarized in Supplementary Table 4, 5.

[(vacancy)Sb] sublayer (Supplementary Fig. 13), leading an overall weakening the bond within the [(Zn,vacancy)Sb]. Secondly, polyatomic-central-shared charges in Sr(Cu,Ag)Sb almost disappear in the Sr$_2$ZnSb$_2$. These variations in charge density confirm the doping of aliovalent Zn and the concomitant introduction of vacancies in the Sr(Cu,Ag,Zn)Sb system bringing strong modification to the atomic bonds.

To quantify the changes in the charge state of each atom in Sr(Cu,Ag,Zn)Sb, Bader charge analysis was further performed and the results are summarized in Table 2. The Sr sublattice in all three compounds shows net charge transfer to the [(Cu,Ag,Zn)Sb] sublattice, further confirming the ionic bonding nature between these two sublattices. These transferred charges are shared by the Cu(Ag) and Sb atoms in Sr(Cu,Ag)Sb. In contrast, these transferred charges within the [ZnSb] sublattice mainly occupy the orbitals around the Sb atoms. More importantly, we find that the charge transfer in SrAgSb (1.370 e) is slightly larger than that in SrCuSb (1.345 e) but it becomes much smaller in Sr$_2$ZnSb$_2$ (1.283 e). The experimental XPS spectra in Fig. 4j clearly establish this apparent change in the amount of transferred charge. Therefore, it is expected that the interlayer ionic bonding is weakened in Sr$_2$ZnSb$_2$.

**Loosely bonded atoms and large anisotropic ADPs**

To verify the theoretical analysis of atomic bonds in Fig. 4a–j, anisotropic ADPs were extracted from the NPD patterns, and the results for the 300 K data are shown in Fig. 4k. SrAgSb shows similar or even slightly smaller ADPs value than SrCuSb, confirming the slightly enhancement of atomic bonding strength in SrAgSb. For the Sr$_2$ZnSb$_2$ compound, the ADPs of the Sr atom exhibits a significant increase in the *ab*-plane ($B_{11}$) as well as along the *c*-axis ($B_{33}$), while Zn and Sb atoms exhibit much larger thermal vibrations along *c*-axis in comparison with Sr(Cu,Ag)Sb. These large changes in the anisotropic ADPs become clearer in the schematic crystal structures (inset of Fig. 4k), which further corroborate the softening and enhanced anharmonicity of the lattice due to the introduction of vacancies. Furthermore, the larger anisotropy of ADPs in the [ZnSb] sublattice demonstrates that

the interlayer bonding presents a larger weakening than the intralayer bonding in Sr$_2$ZnSb$_2$.

## Discussion

Both bulk properties and INS spectra demonstrate lattice softening phenomena in SrAgSb and Sr$_2$ZnSb$_2$ compared with SrCuSb. However, deep analyses of the INS, NPD, and theoretical simulations reveal different physical origins. Condensed matter theory states that phonon group velocity is determined by the atomic mass and bonding strength. In SrAgSb, the lattice softening is attributed to the heavier Ag, which mainly affects the Ag-dominated phonon branches but leaves the atomic bonds intact (Fig. 3b,d,e,h). On the other hand, the softening in Sr$_2$ZnSb$_2$ is due to a large number of vacancies introduced through the substitution of aliovalent Zn for Cu, resulting in the weakening of the interlayer and intralayer bonding. This induces a strong softening of the entire lattice (Fig. 3c–e, i) and makes all the atoms easily shaken by thermal fluctuation (Fig. 4k). Furthermore, the phonon anharmonicity, evaluated by the Grüneisen parameter in Table 1, is also significantly enhanced, manifesting as the overall phonon broadening Fig. 3 and, more importantly, bringing a larger phonon scattering phase space over the whole energy. As shown in Supplementary Fig. 16 and Ref. 39, the three-phonon scattering rate is larger than the four-phonon scattering in all three compounds. However, the replacement of Ag for Cu brings a strong enhancement in the scattering rate only between ~4 and ~9 meV, which should be caused by the strong softening of the low-lying optical phonon due to the heavier atomic mass of Ag. On the other hand, Sr$_2$ZnSb$_2$ presents large increases in both three- and four- phonon scattering rates over the whole energy range in comparison with both SrCuSb and SrAgSb, agreeing well with results from neutron scattering data and atomic bonding analyses. In short, the strongly enhanced phonon softening and anharmonicity, together with phonon-point defect scatterings, endue the Sr$_2$ZnSb$_2$ compound with a much smaller lattice thermal conductivity.

It is worth stressing that the softening in Sr$_2$ZnSb$_2$ happens to the entire lattice except the modes above ~20 meV. This differs from rattler-like compounds, where only part of the atoms in the lattice are weakly bonded, such as TlCuHfSe$_3$, TlInTe$_2$, and Cu$_{12}$Sb$_3$Sb$_{13}$ tetrahedrites[14,44–46]. In the latter cases, low-lying rattling modes are constrained in a narrow energy region and merely intrinsically function as resonant phonon scattering centers. However, the overall softening of the entire Sr$_2$ZnSb$_2$ lattice is largely attributed to its complex hierarchical atomic bonding within and between different sublattices owing to the introduction of vacancies. It can be easily explained using the Zintl concept in Sr$_2$ZnSb$_2$ that the reduction in valence orbitals due to the introduction of vacancies in the [ZnSb] sublattice reduces the space to accommodate the charges transferred from the Sr sublattice, thus weakening the bond strengths. In addition, we would like to point out that the change in the charge transfers causes a sharp drop in the free charge carrier concentration of Sr$_2$ZnSb$_2$ (~4.3 × 10$^{18}$ cm$^{-3}$ at room temperature) in comparison to the SrCuSb and SrAgSb counterparts (1.2 × 10$^{20}$ cm$^{-3}$ and ~4.5 × 10$^{19}$ cm$^{-3}$). It is reasonable to conclude that this lattice softening in Sr$_2$ZnSb$_2$ is also different from the well-studied vacancy-defective SnTe-AgSbTe$_2$, Nb$_{0.8+x}$CoSb, or Pr$_{3-x}$Te$_4$ compounds, where the softening behavior is essentially associated with the free charge carrier concentration and higher carrier concentration is required to lower the phonon group velocities there[47].

In addition, the vacancies in the SnTe-AgSbTe$_2$ compounds were considered to simultaneously play a phonon-vacancy scattering role. Unlike traditional point defect scattering, the vacancy scattering only considers the mass-fluctuation scattering but ignores strain-fluctuation scattering[28,29,48]. However, our high-resolution NPD patterns indicate that Sr$_2$ZnSb$_2$ has a much stronger strain field as it exhibits a larger Bragg peak width than Sr(Cu,Ag)Sb (Supplementary Fig. 17). Therefore, the vacancies in Sr$_2$ZnSb$_2$ should also play a role in

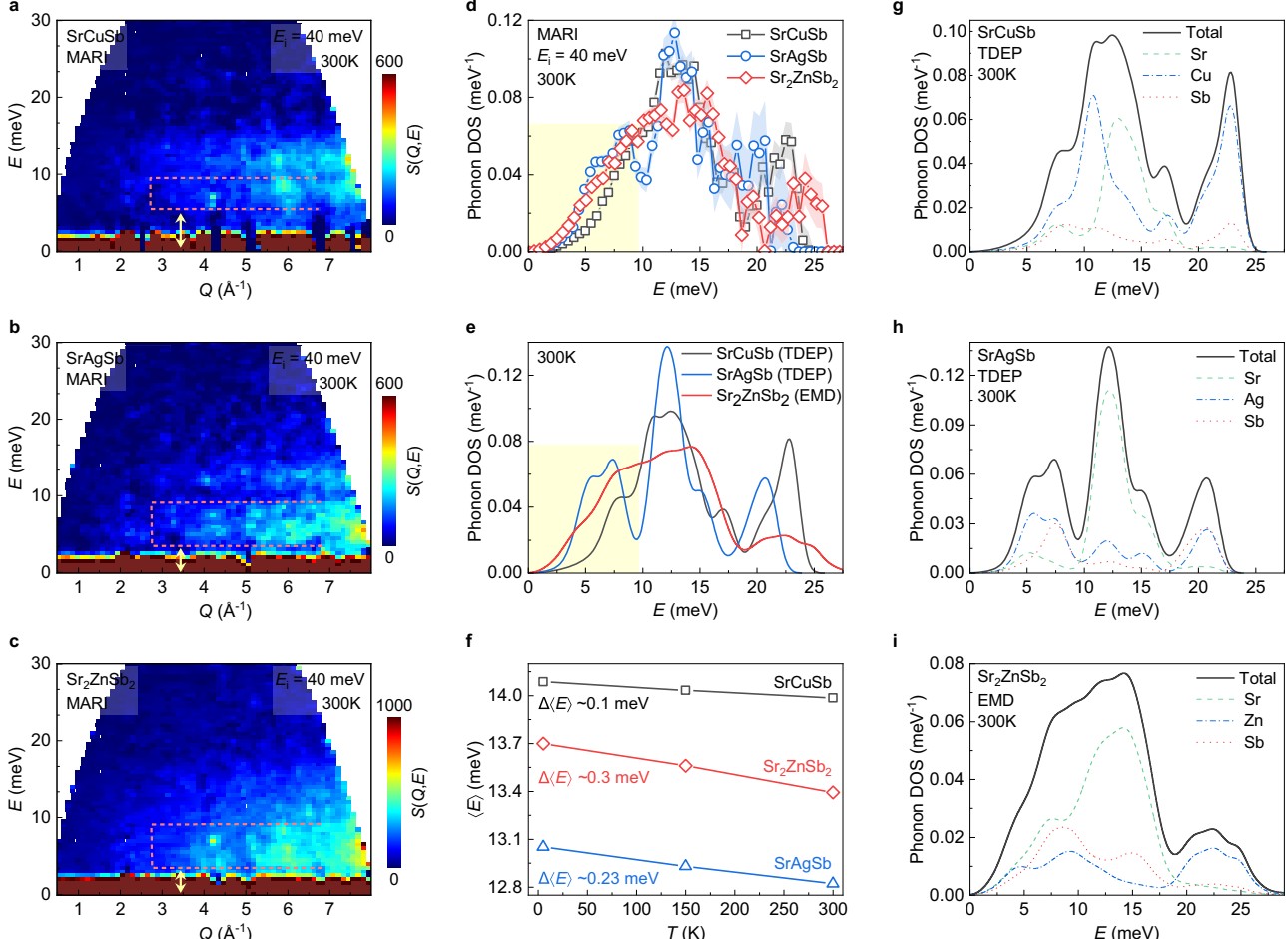

**Fig. 3 | Strong phonon softening and anharmonicity. a–c** Experimental dynamical structure factor, $S(Q,E)$, measured with INS spectrometer, MARI, for the Sr(Cu,Ag,Zn)Sb compounds at 300 K with an incident energy of $E_i = 40$ meV. The pink dashed lines are used to guide the identification of the low-lying flat phonon modes comprising the top of acoustic phonons and low-energy optical phonons (Supplementary Fig. 9). The faint gap between the low- and intermediate-energy flat bands delineates the upper limit of this area, while the top endpoint of acoustic phonon dispersion streaks defines the bottom limit. The energy gap between the bottom limit and the elastic line is marked with yellow arrow. **d** Neutron-weighted phonon DOSs derived from the experimental data in **a–c**. Shaded error bars represent one standard deviation from the phonon DOS calculation using GetDOS[57]. **e** Neutron-weighted phonon DOSs, obtained from temperature dependent effective potential, TDEP, and equilibrium molecular dynamics, EMD, simulations convoluted with instrument resolution at 300 K. The phonon DOSs without considering the different neutron scattering cross sections as well as the simulation results at 0 K or 100 K are provided in Supplementary Figs. 10,11. The yellow-shaded rectangles in **d**, **e** marks softening of the lattice dynamics in Sr(Ag,Zn)Sb compounds. **f** Average phonon energy, $\langle E \rangle = \int E g(E) dE$, calculated from the experimental phonon DOSs at 5, 150, and 300 K (Supplementary Fig. 12). The average phonon energy of Sr$_2$ZnSb$_2$ shows the largest drop with temperature. **g–i** Total and partial neutron-weighted phonon DOSs obtained from TDEP and EMD simulations at 300 K convoluted with instrument resolution. Source data are provided as a Source Data file.

phonon-point defect scattering in addition to softening the phonon group velocities. To better understand the scattering mechanisms, we analyze the lattice thermal conductivity SrAgSb and Sr$_2$ZnSb$_2$ using the Debye-Callaway model (Supplementary Note 4). When we only considered the phonon scattering, as shown in Supplementary Fig. 18, the simulated lattice thermal conductivity of Sr$_2$ZnSb$_2$ is lower than that of SrAgSb, indicating that lattice softening contributes to the low lattice thermal conductivity of Sr$_2$ZnSb$_2$. In addition, vacancy-related scatterings, including the enhanced phonon anharmonicity and phonon-point defect scatterings, are also important to achieve ultralow lattice thermal conductivity with a weak temperature dependence for Sr$_2$ZnSb$_2$.

Intriguingly, recent studies of the EuCuSb-Eu$_2$SnSb$_2$ alloys revealed that Young's, bulk and shear moduli can be continuously tuned on the vacancy concentration[49]. This means that the atomic bonding and the lattice anharmonicity in Zintl-type thermoelectric materials can be continuously and artificially tailored. Previous studies on the lattice imperfections mainly emphasized their phonon-defect

scattering effect in suppressing lattice thermal conductivity. Herein, the experimental and theoretical studies of the Zintl-type Sr(Cu,Ag,Zn) Sb compounds reveal that the role of vacancies is more complex. This is also supported by a recent theoretical study on the lattice dynamics in the vacancy-defective Zr$_{0.88}$NiBi half Heusler compound, which inferred that the vacancies could induce local soft bonds in addition to acting as point-defect scattering[50]. Not limited to the vacancies, it was also found that aliovalent doping in half Heusler compounds could induce softening of optical phonons and avoid crossing of acoustic-optical phonon branches[51,52], while inhomogeneous internal strain fields might be used to modify phonon propagation speed[53]. Further study of lattice defects from a broader perspective is expected to reform the understanding of their roles in solid condensed matter[20]. In summary, the insights gained into the complex lattice dynamics of the vacancy-defective Zintl compounds in this study highlight the potential to exploit defect engineering to artificially tailor lattice anharmonicity to manipulate lattice thermal conductivity in the design of energy conversion materials.

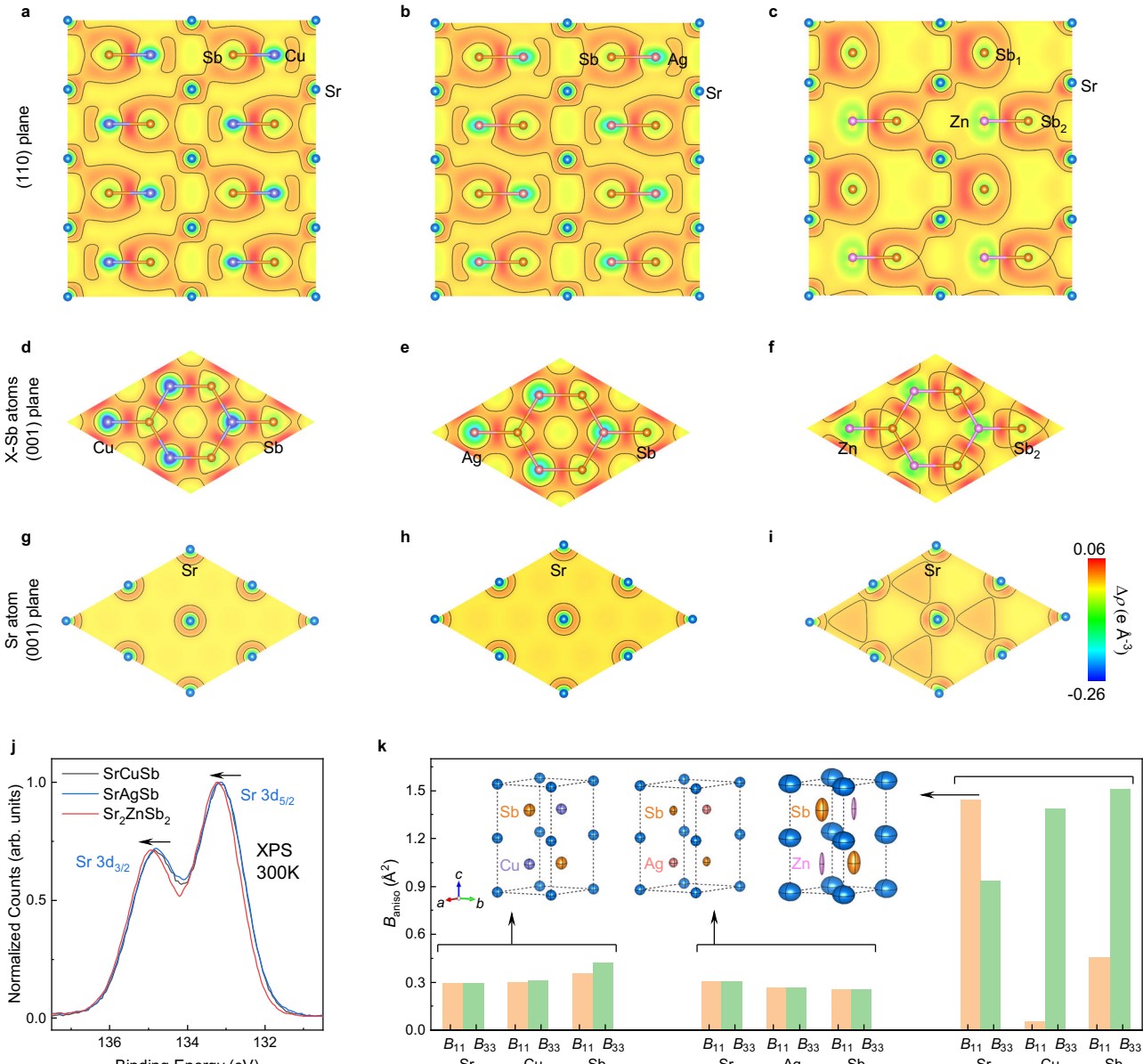

**Fig. 4 | Modified charge distribution and atomic bonding. a–i** Charge density differences, $\Delta\rho$, in Sr(Cu,Ag,Zn)Sb compounds projected in the **a–c** (110)-plane, **d–f** (001) plane of the [X-Sb] layers (X = Cu, Ag, Zn), and **g–i** (001) plane of the Sr layers, visualized with VESTA[67]. (See Supplementary Fig. 13 for the charge density difference in the [(Cu,Ag,Zn)Sb] layers with Zn vacancies.) Different from the experimental determined random distribution in Fig. 1, which was also used for the lattice dynamic simulation in Fig. 3e, i, a $3 \times 3 \times 2$ supercell with ordered vacancies was used for $Sr_2ZnSb_2$ here (Methods). **j** High resolution X-ray photoelectron spectroscopy, XPS, spectra of the $3d_{3/2}$ and $3d_{5/2}$ lines of Sr. The $3d_{3/2}$ (134.97 eV)

and $3d_{5/2}$ (133.21 eV) of Sr element in $Sr_2ZnSb_2$ show blue shift by 0.12 eV in comparison with that (134.85 eV) and (133.09 eV) in SrCuSb, marked with the arrows. More results from XPS are given in Supplementary Fig. 14 and Supplementary Table 6. **k** Anisotropic ADPs, $B_{aniso}$, of the Sr(Cu,Ag,Zn)Sb compounds as determined from the Rietveld refinement of the NPD patterns at 300 K. Insets in **k** show the schematic crystal structures with ADP ellipsoids (for easy comparison, the sizes of the ADP ellipsoids are doubled relative to the atomic distance). The anisotropic ADPs information from 10 K to 300 K are summarized in Supplementary Fig. 15 and Supplementary Tables 7. Source data are provided as a Source Data file.

## Methods

### Sample synthesis

Strontium (Sr, 99.5%, chunk, Aladdin), copper (Cu, 99.99%, wire, ZhongNuo Advanced Material), zinc (Zn, 99.999%, shots, Zhong-Nuo Advanced Material), silver (Ag, 99.99%, pieces, ZhongNuo Advanced Material), and antimony (Sb, 99.999%, shots, Zhong-Nuo Advanced Material) were weighed according to the stoichiometry of Sr2ZnSb2, SrCuSb, and SrAgSb in the glove box under an argon atmosphere with oxygen level below 1.0 ppm. The raw elements (5 g) and grinding balls (16 g) were sealed in a stainless-steel jar (150 mL), and then loaded into the high-energy ball milling machine (SPEX 8000 M). After ball milling for 10 h,

the obtained powders were sintered into bulks with a diameter of 12.7 mm by the spark plasma sintering equipment (LABOX-100, Sinterland) at 973 K for 5 min under a pressure of 6.3 KN. The sample qualities were checked with in-house and synchrotron X-ray diffraction (XRD and SXRD) measurements, which was further confirmed with following NPD measurements.

### Physical properties measurements

**Sound velocity.** The longitudinal and shear components of the sound velocity, $v_l$ and $v_s$, were measured by the sound velocity measurement system, which was grouped by an ultrasonic pulse receiver (Olympus) and an oscilloscope (Tektronix).

**Table 2 | Bader charges analysis on the Zintl-type Sr(Cu,Ag,Zn)Sb compounds**

| Atom | SrCuSb | SrAgSb | $Sr_2ZnSb_2$ |
|---|---|---|---|
| Sr (e) | −1.345 | −1.370 | −1.283 |
| Cu/Ag/Zn (e) | 0.329 | 0.471 | 0.033 |
| Sb (e) | 1.016 | 0.899 | Sb1/Sb2(1.600/0.933) |

Negative values in the table represent the number of electrons lost, and positive values represent the number of electrons gained.

**Heat capacity.** Heat capacity measurement was conducted from 2 to 200 K using the Heat Capacity Option of the Physical Property Measurement System (PPMS, Quantum Design). The addenda heat capacity with N-grease was measured as background in the first step. Under the same conditions, the total heat capacity of Sr(Cu,Ag,Zn)Sb samples (approximately 4.5 mg of each) plus N-grease was measured. Finally, the sample heat capacity was estimated by subtracting the addenda heat capacity from the total value.

**Thermal conductivity.** The total thermal conductivity, $\kappa_{tot}$, was measured using the four-probe lead configuration method based on the Thermal Transport Options (TTO) of PPMS. The gold-plated copper bar purchased from Quantum Design was used as leads during the measurement. The electronic conductivity, $\sigma$, of the material was measured with the Resistivity Option of PPMS. Then, the lattice thermal conductivity, $\kappa_{lat}$, was obtained using $\kappa_{lat} = \kappa_{tot} - \kappa_e$. The carrier contribution, $\kappa_e$, was estimated using the Wiedemann-Franz relationship of $\kappa_e = L\sigma T$, where $L$ is the Lorenz number and calculated using the Seebeck coefficients with a single parabolic band (SPB) model (see Supplementary Fig. 3).

**Electrical properties.** The room-temperature electrical conductivity, $\sigma$, and carrier concentration, $n_H$, were measured using the van der Pauw method. The Hall mobility, $\mu_H$, were calculated using $\mu_H = \sigma/(n_H \times e)$ where $e$ is the electron charge. The room-temperature band gaps were estimated through the optical diffuse reflectance spectra, which was measured using a FTIR system (Thermo Fisher Nicolet iS50). The results are summarized in Supplementary Table 2.

**X-ray photoelectron spectroscopy.** The valence state of the Sr, Cu, Ag, Sb was collected with x-ray photoelectron spectroscopy (ESCALAB 250X, Thermo-Fisher) using Al(Kα) radiation (1486.6 eV).

## Characterization of crystal structures

**X-ray powder diffraction.** In house XRD patterns of Sr(Cu,Ag,Zn)Sb were measured at room temperature on a Rigaku MiniFlex powder diffractometer (Supplementary Fig. 1). Diffraction peaks are well-indexed to the ZrBeSi-type structure ($P6_3/mmc$), and no impurity phase is observed. The SXRD measurement were performed at 300 K on the Powder Diffraction beamline at the Australian Synchrotron. The wavelength was determined as 0.727464 Å by refining the reference sample of $LaB_6$. ~5 mg powder sample was finely and uniformly ground and then loaded into a rotating borosilicate capillary with a diameter of ~0.3 mm. Each pattern was collected for 10 minutes to ensure good statistics.

**Neutron powder diffraction.** NPD was performed on the Super High Resolution Powder Diffractometer, SuperHRPD, at the Materials and Life Science Experimental Facility (MLF), the Japan Proton Accelerator Research Complex (J-PARC). Approximately 5 grams samples were ground into powder in agate mortar and then placed in a vanadium can and the neutron time-of-flight data. The NPD patterns were collected over the temperature range of ~10 K to 300 K. The high resolution of SuperHRPD ($\Delta d/d = 0.03$–$0.15\%$) allows for an accurate determination

of the crystal structure as well as the ADPs. In addition, a further measurement was performed on the High Resolution Powder Diffractometer, ECHIDNA, at the Australian Organization of Nuclear Science and Technology, confirming the quality of the data analysis. Rietveld refinements for NPD and (S)XRD patterns were conducted using Z-Rietveld software[54] and FullProf suite[55].

## Inelastic neutron scattering measurements

The INS measurements were performed using the time-of-flight spectrometer MARI at the ISIS Neutron and Muon Source, UK. The experiment was conducted on 13 grams samples with an incident neutron energy, $E_i$, of 40 meV at temperatures 5, 150, and 300 K with a top loading closed cycle refrigerator. Thin-walled aluminum sample cans were used to reduce the multi-scatterings. The collected spectra were normalized with the incident flux and sample masses. In addition, the spectra of empty can were also collected under the same instrumental configuration and temperature points. The powder-average dynamic structure factors, $S(Q,E)$, were visualized using the DAVE program[56]. Finally, the neutron-weighted phonon DOSs were calculated by integrating the $S(Q,E)$ over $Q$ from 0.5 to 8.0 Å$^{-1}$ after removing the empty can background, multi-phonon and multi-scattering signals using the GetDOS program[57,58].

## First-principles calculations

Phonon simulations and charge density differences of SrCuSb, SrAgSb and $Sr_2ZnSb_2$ were performed in the framework of density functional theory (DFT) as implemented in the Vienna Ab-initio Simulation Packages (VASP)[59]. We used the projector-augmented wave (PAW) pseudopotentials to represent core electrons[60]. The Perdew-Burke-Ernzerhof (PBE) of generalized gradient approximation (GGA-PBE)[61] was applied as exchange-correlation functional for SrCuSb and SrAgSb, while PBEsol[62] was adopted for $Sr_2ZnSb_2$ which could more reasonably reproduce the lattice parameters extracted from the experimental data. We used a kinetic energy cutoff of 400 eV to truncate the plane wave basis set in all DFT calculations. We used Monkhorst-Pack k-point meshes of $9 \times 9 \times 4$ and $3 \times 3 \times 2$, respectively, for the primitive cell and the supercell of SrCuSb, SrAgSb, and $Sr_2ZnSb_2$. During structural relaxation, atomic positions were optimized until atomic forces were smaller than 1 meV Å$^{-1}$ for lattice parameters obtained from our neutron scattering experiment at 10 K. Phonon dispersions and DOS at 0 K were calculated using the small displacement method as implemented in the Phonopy package[63]. We used $3 \times 3 \times 2$ supercells of the primitive cells of SrAgSb, SrCuSb, and $Sr_2ZnSb_2$ and a displacement amplitude of 0.01 Å in all cases. A vacancy-order supercell was adopted for $Sr_2ZnSb_2$ to demonstrate the influence of vacancies on charge density distribution and atomic bonding.

We used the temperature-dependent effective potential (TDEP) method to extract the renormalized second-order force constants[64]. Ab initio molecular dynamics (AIMD) with Nose−Hoover thermostat were performed at 300 K with the experimental lattices. The duration of AIMD simulations was 12 ps with a 4 fs time step, and 50 configurations were selected with the same interval as reference training structures. The remaining AIMD parameters were kept identical to those of the harmonic phonon calculations.

The TDEP approach exploits the symmetry of the lattice to reduce computational workload and improve the numerical accuracy during the renormalization process. This approach becomes genuinely disadvantageous in dealing with the Zn-compound, where the randomly distributed vacancies break the symmetry. On the other hand, the autocorrelation function of velocities based on the equilibrium molecular dynamics (EMD) simulations allows one to simultaneously account for full anharmonicity and occupancy disorder. Therefore, while the lattice dynamics of SrCuSb and SrAgSb were simulated with TDEP, the moment tensor potential (MTP) of $Sr_2ZnSb_2$[39] were built

from EMD simulations under the NVE ensemble using the LAMMPS package[65] at 100 and 300 K. We used a $4 \times 4 \times 3$ supercell (4320 atoms) of the general special quasi-random structure (SQS)[66] with the temperature-dependent experimental lattice and the duration of each EMD simulation was 4 ns with a 4 fs timestep. Before data collection, all systems were initially relaxed by minimizing the potential energy and then equilibrated for 400 ps using the NVT ensemble. The atomic velocities and positions were collected every 10 steps that were subsequently Fourier transformed to evaluate the power spectra.

### Reporting summary

Further information on research design is available in the Nature Portfolio Reporting Summary linked to this article.

## Data availability

All data that support the findings of this study are provided within the paper and its Supplementary Information. All additional information is available from the corresponding authors upon reasonable request. Source data underlying Figs. 1g, h, 2a–g, 3a–i, and 4j are provided with this paper. Source data are provided with this paper.

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

## Acknowledgements

J.M. are grateful for the National Key Research and Development Program of China (grant no. 2022YFA1402702). Q.R. thanks the Guangdong Basic and Applied Basic Research Foundation (grant no. 2021B1515140014) and the National Natural Science Foundation of China (grant no. 52101236). J.M. also thanks the funding support by the National Natural Science Foundation of China (grant no. U2032213). Q.Z. thanks to the National Natural Science Foundation of China (grant no. 51971081 and 52172194), the Shenzhen Science and Technology Program (KQTD20200820113045081 and RCJC20210609103733073). C.C. thanks the National Natural Science Foundation of China (grant no. 52001339). C.W. and Y.C. are grateful for the financial support of the Research Grants Council of Hong Kong (C7002-22Y) and the research computing facilities offered by ITS, HKU. Z.Q. thanks to support from CAS Project for Young Scientists in Basic Research (grant no. YSBR-084) and the National Natural Science Foundation of China (grant no. U2032213 and 12374129). A portion of this work was supported by the High Magnetic Field Laboratory of Anhui Province. Z.C. thanks Australia Research Council for support (DP210101436). Q.R. and X.T. thank the support from Guangdong Provincial Key Laboratory of Extreme Conditions (grant no. 2023B1212010002). Q.R. thanks Gaoting Lin, Hanjie Guo, and Jian Lv for their assistance in the measurements of the electrical transport properties. We all acknowledge the beam time granted by ISIS (experiment RB2010368), J-PARC (proposal no. 2019B0215), ANSTO (proposal no. MI8479), and the technical support from Q. Gu during the experiments at the Australian Synchrotron.

## Author contributions

Q.R., C.C., Q.Z., and J.M. conceived the project. C.C. and Q.Z. prepared the samples. J.Z. and C.Z. carried out the transport properties measurements. Q.R. and J.M. performed the NPD and INS measurements with input from M.D.L., S.T., C.-W.W., and G.W. J.Z. and Q.R. carried out Rietveld refinement of the NPD and (S)XRD data. J.Z. and Q.R. analyzed the INS data with help from L.L., X.T. and J.M. Y.J. and M.W. took the XPS experiment and the results were by J.Z., M.S. and Q.R. C.C., Q.Z., M.H., and Z.Q. conducted in house XRD, and J.W. and Z.C. conducted the SXRD experiment and the results were analyzed by J.Z. and Q.R. C.W. and Y.C. carried out the first-principles calculations. J.Z. and Q.R. explained the results and drafted the original manuscript. Q.R. and J.M. supervised the project. All the authors edited and finalized the manuscript.

## Competing interests

The authors declare no competing interests.
