## [Peer Review File · Nature Communications]

Vacancies tailoring lattice anharmonicity of Zintl-type thermoelectricsREVIEWER COMMENTS

Reviewer #1 (Remarks to the Author):

See attached review file.

Reviewer #2 (Remarks to the Author):

In this manuscript, Zhu et al. investigated the impact of vacancies on the low lattice thermal conductivity in Zintl Sr(Cu, Ag, Zn)Sb compounds. They observed a significant reduction in intrinsic lattice thermal conductivity in Sr₂ZnSb₂, attributing it to extensive vacancies that enhance lattice phonon softening, lattice anharmonicity, and defect scattering. The experimental comparison, lattice dynamic measurements, and theoretical exploration presented in this study contribute to a deeper understanding of the role of vacancies in thermoelectric materials. Therefore, I would like to recommend this manuscript for publication after addressing the following concerns.

1. In Figure S2, fluctuations in the measured electrical resistivity for SrCuSb and SrAgSb are evident. The authors should provide an explanation for these fluctuations. Additionally, the mention of estimating the Lorenz number using the SPB model raises questions about whether the Seebeck coefficient and carrier concentration were measured. If so, these data should be provided; otherwise, the calculation formula for the Lorenz number should be provided.
2. The statement, "This suggests that the origins of the lattice softening in SrCuSb and Sr₂ZnSb₂ are different," might be a typographical error. It appears that "SrAgSb" should be mentioned instead of "SrCuSb."
3. Where are the in-house XRD patterns?
4. The authors should explain why TDEP calculations for Cu- and Ag- samples match well with the experimental data, while EMD calculation for Zn-compound is deemed better.
5. Considering the general challenges in synthesizing pure-phase Zintl materials, it is recommended that more details on the synthesis process should be provided.
6. The statement, "The above two behaviors would undermine the electrical conductivity as demonstrated in Supplementary Fig. 2a,b," requires further clarification. The authors may need to provide a more detailed explanation of this statement, taking into account the differences in band structure and potential cationic vacancies in these materials. Additional context may be necessary.

Reviewer #3 (Remarks to the Author):

Zhu et al. investigate the roles of vacancies in thermal transport, using exemplary ZrBeSi-type Zintl phases for their analysis. Zintl phases, particularly those that adopt the ZrBeSi structure type, have recently attracted interest for thermoelectric applications; as a result, elucidating mechanistic roles of lattice dynamics on thermal properties is timely.

However, the reduction in lattice thermal conductivity due to vacancy-induced lattice softening is a well-documented phenomenon, as demonstrated in numerous highly-defective systems (see, e.g., Slade et al., *Joule* 5, 2021). The phenomenon has been cited in the context of thermoelectric performance in ZrBeSi-type phases also (see, e.g., Chanakian et al., *Angew. Chem.* 62, 2023). Moreover, shortcomings in the presented analysis leave room

for a more complete analysis of phonon dynamics in ZrBeSi-type Zintl phases.

The paper is interesting, and the neutron diffraction experiments on the Zintl phases are indeed appreciated. Unfortunately, I do not think it provides results of exceptional quality or novelty to merit publication in Nature Communications. Please see below for my reasoning.

Major points:

1. The absence of long-range ordering in Sr₂ZnSb₂ is stated in the text, but not supported by the provided data. Since this motivated the structural refinement and subsequent simulations (as indicated in lines 106-109), please provide an explanation for the Zn vacancy ordering.
2. Regarding the statement on lines 208-209: "Only the Ag-related phonon modes around 10 meV exhibit obviously softening in SrAgSb while the energy for other modes does make large variations.": the term "softening" in this context does not make sense, as it is a term used to describe the lowering of the phonon group velocity (see, e.g., Hanus et al., Adv. Mater. 31, 2019). The authors even implied that the term "softening" refers to a reduction in the group velocity when comparing the Debye temperatures in line 147. How is the phonon DOS related to the group velocity? If a more general definition of "softening" is adopted in this study, please state that.
3. The reduction in lattice thermal conductivity in Sr₂ZnSb₂, compared to SrCuSb and SrAgSb, is mainly attributed to lattice softening induced by vacancies in this study. However, the role of defect scattering, which can significantly affect the lattice thermal conductivity as shown in papers cited by the authors (e.g., Refs 29 and 30), is not fully investigated in this study. Further analysis should be provided to thoroughly elucidate the mechanisms underlying the low lattice thermal conductivity in the 2-1-2 phase.

Minor points:

1. Regarding the statement on lines 156-157: "It is easy to read from Fig. 2g that the optical phonon modes in both Ag and Cu compounds shift towards lower energy.": it is unclear how it can be understood that optical phonon modes have lower frequencies in SrCuSb and SrAgSb. Please provide a short explanation.
2. Why is the PBE functional used for SrCuSb and SrAgSb, but PBEsol for Sr₂ZnSb₂? Would the usage of different DFT functionals lead to inconsistent results? Please provide an explanation.
3. Are the phonon DOS in Fig. 3e calculated using two different methods? If so, is this because the random distribution of Zn vacancies in Sr₂ZnSb₂ breaks periodicity? Please explain why different methods are used.
4. Do the Gruneisen parameters for the three compounds listed in Table 1 match with DFT-calculated values? Such an analysis can strengthen the argument that Sr₂ZnSb₂ exhibits stronger anharmonicity than SrCuSb and SrAgSb, as well as provide further credence

towards the computational results as a whole.

Spelling errors (which did not affect the decision):

1. Line 106: "Wort" is misspelled.
2. Line 409: "Ernzerh" is misspelled.

In this work, the authors synthesize Sr_2ZnSb_2 in which Zn-induced vacancies have previously been predicted to enhance anharmonicity thereby lowering lattice thermal conductivity using first-principles method (*Chem. Mater.* 2022, 34, 7837–7844). Their measurements indicate low lattice thermal conductivity that agrees with their previous theoretical predictions. The authors further invoke various spectroscopic techniques, supported by some additional first-principles calculations, to describe lattice softening and increased anharmonicity. The studies performed and their analysis appear sound and their conclusions well-supported. The notable merit is that this constitutes an experimental success as guided and predicted by theory and computation, representing the power of their synergy. I have some relatively minor comments that the authors should address prior to publication.

In Introduction,

1. The authors start discussing Boltzmann transport theory with equation $\kappa_{\text{lat}} = 1/3c_V v^2 \tau$, but this does not represent Boltzmann transport theory. It is just plain kinetic theory of gasses.
2. The authors include ferroelectric instability as something that induces anharmonicity. It is the opposite – anharmonicity induces ferroelectric instability.
3. The authors say “...all of above factors are intrinsic properties,” and it is nearly impossible to artificially modify them.” This conflicts with the whole paper. The whole point of the work is that one could introduce impurities/disorders and such to a sufficient extent such that those intrinsic properties are significantly modified. Anything one does to the parent lattice to an extent that perturbation theory becomes inapplicable is deemed as substantially changing the intrinsic properties.
4. On a similar note, the authors say regarding that point defect scattering, “...originating from mass and strain fluctuations but without disturbing phonon energy and group velocity.” This is again not true. Anything one does to the lattice will change everything, if negligibly. The picture the authors paints is only true under Perturbation theory.

In Results and Discussion, I think the authors should explicitly compare their measured lattice thermal conductivity of Sr_2ZnSb_2 to the previously computed values from *Chem. Mater.* 2022, 34, 7837–7844. Perhaps add it to Fig. 1h as well.

As shown in Supp, the authors calculate experimental Gruneisen parameter using sound velocities. Sound velocity however does not contain anharmonic information. It would be more accurate to calculate this using thermal expansion, which is a function of 3rd-order anharmonicity if they can measure CTE easily.

As for the computation part, there is a little bit more trivial work that the authors could do to make their computation-experiment synergy stronger. The authors have computed T-dependent phonons. They can actually calculate mean-square displacements from the phonons and compare them to experimentally inferred values.

Supplementary Fig. 8c and 8d should be overlaid such that the motion of the imaginary mode is made clear.

Finally, I wish the authors commented on the possibility of doping Sr_2ZnSb_2 as to improve its electronic transport since they are describing the material as a thermoelectric. While low lattice thermal conductivity is nice, it alone will not achieve high thermoelectric efficiency as we all know, especially when the electronic properties of Sr_2ZnSb_2 seem so poor as they stand.

Responses to Reviewers' comments:

Reviewer #1 (Remarks to the Author):

In this work, the authors synthesize Sr₂ZnSb₂ in which Zn-induced vacancies have previously been predicted to enhance anharmonicity thereby lowering lattice thermal conductivity using first-principles method (Chem. Mater. 2022, 34, 7837–7844). Their measurements indicate low lattice thermal conductivity that agrees with their previous theoretical predictions. The authors further invoke various spectroscopic techniques, supported by some additional first-principles calculations, to describe lattice softening and increased anharmonicity. The studies performed and their analysis appear sound and their conclusions well-supported. The notable merit is that this constitutes an experimental success as guided and predicted by theory and computation, representing the power of their synergy. I have some relatively minor comments that the authors should address prior to publication.

Response: We thank the Reviewer for the positive comments and helpful suggestions on our work. The comments and suggestions have helped us to improve the manuscript. The corresponding responses and changes are as follows.

Comment 1. The authors start discussing Boltzmann transport theory with equation $\kappa_{\text{lat}} = 1/3c_V v^2 \tau$, but this does not represent Boltzmann transport theory. It is just plain kinetic theory of gasses.

Response: We thank the Reviewer for pointing out this wrong description of the formula. It is actually the Debye-Callaway model of free phonon gases. In the revised manuscript, we replaced the “Boltzmann transport theory” with the expression of the “classical Debye-Callaway model of free phonon gases”: “Within the classical Debye-Callaway model of free phonon gases, the phonon-dominated lattice thermal conductivity can be simply written as $\kappa_{\text{lat}} = 1/3c_V v^2 \tau$, where c_V is the lattice heat capacity, v is the phonon group velocity and τ is the phonon relaxation time or lifetime.”

Comment 2. The authors include ferroelectric instability as something that induces anharmonicity. It is the opposite – anharmonicity induces ferroelectric instability.

Response: We agree with the Reviewer's comment that anharmonicity induces ferroelectric instability. To make the description more accurate, we modified the sentence as: “Furthermore, stereochemically active lone pair of ns^2 electrons¹⁰, weak chemical bonding^{11,12}, resonant bonding¹³, rattler atomic vibrations^{14,15}, and four-phonon Fermi resonance¹⁷ always induce strong phonon anharmonicity and introduce Umklapp scattering, which can significantly shorten the phonon lifetime⁸.”

Comment 3. The authors say "...all of above factors are intrinsic properties," and it is nearly impossible to artificially modify them." This conflicts with the whole paper. The whole point of the work is that one could introduce impurities/disorders and such to a sufficient extent such that those intrinsic properties are significantly modified. Anything one does to the parent lattice to an extent that perturbation theory becomes inapplicable is deemed as substantially changing the intrinsic properties.

Response: We appreciate the Reviewer for this suggestion. To make a more accurate discussion, we modified this sentence in the revised manuscript: "All of the above factors are intrinsic properties and mainly depend on a given material's crystal symmetry and chemical components. However, it is difficult to modify them artificially within the valid range of perturbation theory."

Comment 4. On a similar note, the authors say regarding that point defect scattering, "...originating from mass and strain fluctuations but without disturbing phonon energy and group velocity." This is again not true. Anything one does to the lattice will change everything, if negligibly. The picture the authors paints is only true under Perturbation theory.

Response: Following the Reviewer's suggestions, we have made a change to the sentence: "Most of them are traditionally regarded as important phonon scattering mechanisms in thermal transport, originating from mass and strain fluctuations, while their influences on phonon energy and group velocity have been overlooked."

Comment 5. In Results and Discussion, I think the authors should explicitly compare their measured lattice thermal conductivity of Sr₂ZnSb₂ to the previously computed values from Chem. Mater. 2022, 34, 7837–7844. Perhaps add it to Fig. 1h as well.

Response: Thank the Reviewer for this suggestion. We have summarized the lattice thermal conductivity data from different methods, including theoretical simulations, light flash method with laser flash apparatus [*Chem. Mater.* **34**, 7837-7844 (2022)], and steady-state measurements with PPMS as done in this work. As shown in Response Table 1 and Response Fig. 1, all the data from different methods show good agreement.

Response Table 1 | Comparison of the lattice thermal conductivities for Sr(Cu,Ag,Zn)Sb compounds as determined from different methods, including theoretical simulation (marked as Simulated) at 300 K, light flash method with laser flash apparatus (marked as LFA) at 300K, and steady-state measurements with PPMS at 280 K in this work (marked as PPMS). The simulated and FLA data are cited from our previous work of [*Chem. Mater.* **34**, 7837-7844 (2022)].

κ_{lat} (W m ⁻¹ K ⁻¹)	T (K)	SrCuSb	SrAgSb	Sr ₂ ZnSb ₂
Simulated [Chem. Mater.]	300	3.78	-	0.46
LFA [Chem. Mater.]	300	3.25	1.55	0.45
PPMS [this work]	280	3.59	1.53	0.63

Response Fig. 1 | Comparison of the lattice thermal conductivities for Sr(Cu,Ag,Zn)Sb compounds as determined from different methods. The curves over 2 to 280 K were measured using a steady-state method with PPMS [this work]. The point data marked with diamond were obtained from theoretical simulation at 300K [Chem. Mater. 34, 7837-7844 (2022)], and the star data were measured with light flash method using laser flash apparatus at 300K [Chem. Mater. 34, 7837-7844 (2022)].

Comment 6. As shown in Supp, the authors calculate experimental Grüneisen parameter using sound velocities. Sound velocity however does not contain anharmonic information. It would be more accurate to calculate this using thermal expansion, which is a function of 3rd -order anharmonicity if they can measure CTE easily.

Response: We thank the Reviewer for raising this issue, which encourages us to think more about the techniques used to estimate Grüneisen parameters. The Grüneisen parameter, γ , is a measure of the anharmonicity of interactions between atoms or molecules in solids, and microscopically defined as:

$$\gamma = -\frac{\partial \ln \omega_j}{\partial \ln V} = -\frac{V}{\omega_j} \left(\frac{\partial \omega_j}{\partial V} \right) \quad (R1)$$

where V is the volume and ω_j is the frequency of phonon vibration normal mode j . The elastic nonlinearity of condensed matter is related to nonlinear acoustic properties via anharmonic interactions [Tech. Phys. Lett. 30, 91-93 (2004)]. This leaves a possibility to macroscopically estimate the Grüneisen parameter from bulk properties.

Currently, there are two popular methods to macroscopically estimate the Grüneisen parameter. The first one is based on the Debye model and assumes the longitudinal and transverse phonons take the same truncation Debye frequency, ω_D . Then, one can derive the Grüneisen parameter, γ_D , following elasticity theory and is referred as thermodynamic formula:

$$\gamma_D = \frac{3\alpha V B_T}{C_V} \quad (R2)$$

where α is coefficient of thermal expansion (CTE), B_T is isothermal bulk modulus, and C_V

is the heat capacity (the CET method suggested by the reviewer should refer to this formula). The second one was developed by Leont'ev using the propagation velocities of acoustic waves (or phonons), and referred as Leont'ev formula:

$$\gamma_a = \frac{3}{2} \left(\frac{3v_l^2 - 4v_s^2}{v_l^2 + 2v_s^2} \right) \quad (\text{R3})$$

where the v_l and v_s are the longitudinal and shear sound velocities. At the first glance, the sound velocity does not contain anharmonic information. However, a careful study finds that both bulk modulus and CET in the thermodynamic formula (Equ. R2) correlate to the sound velocity:

$$\rho v_l^2 = B_T + \frac{4G}{3}, \rho v_s^2 = G \quad (\text{R4})$$

$$\bar{v}^2 = \frac{3}{2} \frac{C_V}{\alpha \rho V} \quad (\text{R5})$$

Here, G is the shear modulus. \bar{v} is the root-mean-square sound velocity determined as:

$$\bar{v} = \left(\frac{v_l^2 + 2v_s^2}{3} \right)^{1/2} \quad (\text{R6})$$

The relationship between CTE and sound velocity as demonstrated in Equ. R5 and its correctness were discussed by Leont'ev in [*Sov. Phys. Acoust.* **27**, 309 (1981)]. In addition, Belomestnykh demonstrates that the Grüneisen parameter, γ_D and γ_a , as determined from the thermodynamic formula and the Leont'ev formula, are in good agreement in most cases [*Tech. Phys. Lett.* **30**, 91-93 (2004)]. Although some big discrepancies were found, Sanditov et al. found that this could be explained with the different Poisson's ratios [*Tech. Phys.* **56**, 1619-1623 (2011)]. In this work, we used the Leont'ev formula of Eqs. R3 and sound velocity to estimate the Grüneisen parameter, which should be equal to the CTE method (or thermodynamic formula of Equ. R2) as suggested by the Reviewer.

However, it should be noted that both γ_D and γ_a have a pre-assumption that the phonon DOS is simplified as a Debye model. This might bring large inaccuracy in estimating the Grüneisen parameters for SrAgSb and Sr₂ZnSb₂ as these two samples have much lower optical phonons that overlap with the acoustic phonon. Therefore, to response this comment to make the comparison of Grüneisen parameters between the three samples more convincing, rather than take a CTE measurement, we estimate the Grüneisen parameters from the phonon density of states (DOS), $g(E)$, as determined from inelastic neutron scattering measurement. An average Grüneisen parameter, $\bar{\gamma}$, could be obtained with the following formula [*Phys. Rev. B* **80**, 184302 (2009)]:

$$\bar{\gamma} = - \frac{d \ln \langle E \rangle}{d \ln V} \quad (\text{R7})$$

where $\langle E \rangle = \int E g(E) dE$ is the average phonon energy. Combined with the unit cell volume obtained from the refinement of NPD data, $\bar{\gamma}$ of SrCuSb, SrAgSb and Sr₂ZnSb₂ are estimated as 0.769, 1.739 and 2.799, respectively. The Sr₂ZnSb₂ shows an obviously larger value than Sr(Cu,Ag)Sb, corroborating the conclusions of this work that vacancies bring about stronger phonon anharmonicity in Sr₂ZnSb₂. Furthermore, the larger three-phonon and four-phonon scattering rates of Sr₂ZnSb₂ than that of Sr(Cu,Ag)Sb also suggest that the Sr₂ZnSb₂ has a stronger phonon anharmonicity (refer the discussion in Supplementary Fig. 16).

Following this response, the $\bar{\gamma}$ values are summarized in Table 1 in the revised manuscript. In addition, we modified the discussion in the second paragraph of the “Lattice dynamics and strong phonon anharmonicity” section as: “Moreover, the sound velocity analysis yields close Grüneisen parameters for SrCuSb (1.42) and SrAgSb (1.43), but the Sr₂ZnSb₂ gives a larger value of 1.50 (Table 1). This enlarged behavior of the phonon Grüneisen parameter in Sr₂ZnSb₂ is also captured by the average Grüneisen parameters, $\bar{\gamma} = d \ln \langle E \rangle / \ln V$, where $\langle E \rangle = \int E g(E) dE$ is the average phonon energy (Fig. 3f) and V is unit cell volume determined from NPD patterns. Here, the different absolute values of the Grüneisen parameter from these two methods might come from the over-simplified Debye model used in the sound velocity method.”

Comment 7. As for the computation part, there is a little bit more trivial work that the authors could do to make their computation-experiment synergy stronger. The authors have computed T-dependent phonons. They can actually calculate mean-square displacements from the phonons and compare them to experimentally inferred values. Supplementary Fig. 8c and 8d should be overlaid such that the motion of the imaginary mode is made clear.

Response: We appreciate the Reviewer for these two suggestions.

1) For the mean-squared displacements (MSD): The absolute value of MSD value extracted from the Rietveld refinement of diffraction patterns depends on a lot of external factors, especially the instrumental backgrounds and resolution functions. Therefore, a direct comparison between the computation values and experimental values, or between the experimental values from different instruments, is difficult. However, the temperature-dependent trends and the variation trends between different samples should be reliable. Especially, the variation trends between different samples are quite important in this work as one of the main findings is that the Sr₂ZnSb₂ compound has a much larger MSD than the other two compounds as Sr₂ZnSb₂ suffers an overall softening to the entire lattice due to the introduction of a large number of vacancies.

To properly respond to this comment, instead of taking theoretical calculations of the MSD, we rather take more analysis of atomic displacement parameters (ADP, a crystallographic nomenclature) from the neutron powder diffraction patterns collected on the Echidna diffractometer (at ANSTO reactor neutron source in Australia). As summarized in Response Fig. 2a, the overall ADP, B_{ov} , of Sr₂ZnSb₂ is much larger than that of Sr(Cu,Ag)Sb as extracted

from the Rietveld refinements of the Echidna data. This trend is the same as the results from the NPD patterns collected on the SuperHRPD diffractometer (at J-PARC spallation neutron source in Japan) as shown in Response Fig. 2b. The consistency of the ADPs from two different diffractometers confirms that Sr_2ZnSb_2 has a loosely bonded crystal structure, which is different from $\text{Sr}(\text{Cu,Ag})\text{Sb}$.

Response Fig. 2 | Overall isotropic atomic displacements, B_{ov} , for the $\text{Sr}(\text{Cu,Ag,Zn})\text{Sb}$ compounds. The data are extracted the Rietveld refinements of the neutron powder diffraction patterns collected on **a**, Echidna diffractometer at the ANSTO reactor neutron source in Japan and **b**, SuperHRPD diffractometer at the J-PARC spallation neutron source in Australia. Both data show larger B_{ov} for Sr_2ZnSb_2 than that for for $\text{Sr}(\text{Cu,Ag})\text{Sb}$.

2) With respect to the imaginary mode: The Fig. 8c and 8d have been merged together to make it easy to compare the imaginary mode. In the revised Supplementary Information, the original Supplementary Fig. 8 becomes the Supplementary Fig. 9:

Supplementary Fig. 9 | Phonon dispersions from theoretical calculations.

Comment 8. Finally, I wish the authors commented on the possibility of doping Sr_2ZnSb_2 as to improve its electronic transport since they are describing the material as a thermoelectric. While low lattice thermal conductivity is nice, it alone will not achieve high thermoelectric efficiency as we all know, especially when the electronic properties of Sr_2ZnSb_2 seem so poor as they stand.

Response: Thank the reviewer for this suggestion. The ultralow lattice thermal conductivity is a beneficial factor for Sr_2ZnSb_2 to achieve promising thermoelectric properties, however, the

electrical transport properties are not expected due to the low carrier concentration. Some approaches have been tried to optimize the carrier concentration, such as regulating the element content and element doping, but the carrier concentration is hard to be regulated. It is mainly because of the low doping efficiency. We believe that some novel preparation technology, for example the ion/electron implantation, will contribute to improving the carrier concentration in the future.

Reviewer #2 (Remarks to the Author):

In this manuscript, Zhu et al. investigated the impact of vacancies on the low lattice thermal conductivity in Zintl Sr(Cu, Ag, Zn)Sb compounds. They observed a significant reduction in intrinsic lattice thermal conductivity in Sr₂ZnSb₂, attributing it to extensive vacancies that enhance lattice phonon softening, lattice anharmonicity, and defect scattering. The experimental comparison, lattice dynamic measurements, and theoretical exploration presented in this study contribute to a deeper understanding of the role of vacancies in thermoelectric materials. Therefore, I would like to recommend this manuscript for publication after addressing the following concerns.

Response: We are grateful to the Reviewer for the recognition of the high quality and the significance of our manuscript. We also thank the Reviewer for the constructive comments on improving this article. Below are our detailed responses to each of the comments.

Comment 1. In Figure S2, fluctuations in the measured electrical resistivity for SrCuSb and SrAgSb are evident. The authors should provide an explanation for these fluctuations. Additionally, the mention of estimating the Lorenz number using the SPB model raises questions about whether the Seebeck coefficient and carrier concentration were measured. If so, these data should be provided; otherwise, the calculation formula for the Lorenz number should be provided.

Response: Thank the reviewer for pointing out this issue. In the revised manuscript, we have provided the Seebeck coefficient and the Lorenz number curves calculated using the Seebeck coefficient with the SPB model. In addition, we also re-measured the electrical resistivity for SrCuSb and SrAgSb, and the new data demonstrate smooth temperature dependencies. The fluctuations in the old electrical resistivity data might be attributed to the improper selection of current value during the measurements. The new data, especially the lattice thermal conductivities, is quite close to the original data, and it does not affect our discussions and conclusions in this work. The updated data are shown as following:

Supplementary Fig. 3 | Thermal and carrier transport properties of Sr(Cu,Ag,Zn)Sb compounds.

a. Resistivity, ρ . **b.** Carrier conductivity, σ . **c.** Seebeck coefficient, S . **d.** Lorenz number, L . **e.** Temperature variable total thermal conductivity, κ_{tot} . **f.** Electrical thermal conductivity, κ_e . It is noted that the Sr_2ZnSb_2 sample has a much lower electrical conductivity in comparison to the other two samples as shown in (b). According to the Hall carrier concentration, mobility, and band gap measurements (see Supplementary Table 1), this poor electrical conductivity is attributed to the much lower Hall carrier concentration and much smaller mobility.

Comment 2. The statement, "This suggests that the origins of the lattice softening in SrCuSb and Sr₂ZnSb₂ are different," might be a typographical error. It appears that "SrAgSb" should be mentioned instead of "SrCuSb."

Response: Thank the Reviewer for pointing out this typographical error. We have made correction in the manuscript: "This suggests that the origins of the lattice softening in SrAgSb and Sr₂ZnSb₂ are different."

Comment 3. Where are the in-house XRD patterns?

Response: We provided the in-house XRD patterns in Supplementary Fig.1.

Supplementary Fig. 1 | In-house XRD patterns at 300 K. The patterns are presented in the plane distance, d , space. All three samples can be indexed with the hexagonal $P6_3/mmc$ structure but without discernible impurities. The (102), (110) and (004) Bragg peaks are labeled.

Comment 4. The authors should explain why TDEP calculations for Cu- and Ag- samples match well with the experimental data, while EMD calculation for Zn-compound is deemed better.

Response: The temperature dependent effective potential (TDEP) approach exploits the symmetry of the lattice to reduce computational workload and improve the numerical accuracy during the renormalization process. This approach becomes genuinely disadvantageous in dealing with the Zn-compound with the general special quasi-random structure (SQS), where randomly distributed vacancies break the symmetry. Therefore, we calculated the power spectra from the autocorrelation function of velocities based on the equilibrium molecular dynamics (EMD) simulations, which allows us to simultaneously account for full anharmonicity and disorder.

Following this comment, we added an explanation in the Methods: “The TDEP approach exploits the symmetry of the lattice to reduce computational workload and improve the numerical accuracy during the renormalization process. This approach becomes genuinely disadvantageous in dealing with the Zn-compound, where the randomly distributed vacancies break the symmetry. On the other hand, the autocorrelation function of velocities based on the equilibrium molecular dynamics (EMD) simulations allows one to simultaneously account for full anharmonicity and occupancy disorder. Therefore, while the lattice dynamics of SrCuSb and SrAgSb were simulated with TDEP, the moment tensor potential (MTP) of Sr₂ZnSb₂ were built from EMD simulations under the NVE ensemble using the LAMMPS package at 100 and 300 K.”

Comment 5. Considering the general challenges in synthesizing pure-phase Zintl materials, it is recommended that more details on the synthesis process should be provided.

Response: The sample synthesis method is updated in the revised manuscript, and more details are provided: “Strontium (Sr, 99.5%, chunk, Aladdin), copper (Cu, 99.99%, wire, ZhongNuo Advanced Material), zinc (Zn, 99.999%, shots, ZhongNuo Advanced Material), silver (Ag, 99.99%, pieces, ZhongNuo Advanced Material), and antimony (Sb, 99.999%, shots, ZhongNuo Advanced

Material) were weighed according to the stoichiometry of Sr₂ZnSb₂, SrCuSb, and SrAgSb in the glove box under an argon atmosphere with oxygen level below 1.0 ppm. The weighed elements (total 5 g) and stainless balls (total 16 g) were loaded into a stainless-steel jar (150 mL, 8001 Harden Steel Vial) and ball milled continuously for 10 h by a high energy ball mill (SPEX 8000M). After that, the obtained powders were loaded into a graphite die with an inner diameter of 12.7 mm in a glove-box. The dense bulks were obtained by spark plasma sintering (LABOX-100, Sinterland) at 973 K for 5 minutes under an axial pressure of 50 MPa. The sample qualities were checked with in-house and synchrotron X-ray diffraction (XRD and SXRD) measurements, which was further confirmed with following NPD measurements.”

Comment 6. The statement, "The above two behaviors would undermine the electrical conductivity as demonstrated in Supplementary Fig. 2a,b," requires further clarification. The authors may need to provide a more detailed explanation of this statement, taking into account the differences in band structure and potential cationic vacancies in these materials. Additional context may be necessary.

Response: Thank the Reviewer for the comment and suggestion. The main information we would like to convey here by discussing these two changes in the charge densities is that the introduction of a large amount of vacancies in Sr₂ZnSb₂ bring strong modification to the atomic bonds. This change in the atomic bonds was further confirmed by the Bader charge analysis and experimental XPS spectra in next paragraph.

We agree with the Reviewer that the attempt to directly link these changes in charge densities to the poor electrical conductivity of Sr₂ZnSb₂ is somewhat far-fetched, and additional context is necessary. Following this suggestion, we tried to calculate the electronic structures of these three samples. However, there is a technical difficulty to accurately estimate the band gap for the vacancy-defective Sr₂ZnSb₂ sample. The difficulty lies on the construction of the random vacancy distribution model used for the first-principle calculation. Our previous study of the Eu₂ZnSb₂ in [*Sci. Adv.* **7**, eabd6162 (2021)] found that the electronic structure strongly depends on the supercell models used for the electronic structure calculation, and the resultant electronic structure could vary from metal to semiconductor.

Therefore, to properly explain the poor electrical conductivity, we rather performed direct measurements on the Hall carrier concentration, Hall mobility, and band gap. The results are summarized in Supplementary Table 1. While the three compounds have similar band gaps close, the Sr₂ZnSb₂ has much lower Hall carrier concentration and mobility. The lower Hall carrier concentration may correlated to the less transferred charges from the Sr sublattice to the [ZnSb₂] sublattice, and the latter is considered to dominate the electrical transport in the Sr(Cu,Ag,Zn)Sb Zintl phases. Additionally, the intrinsic Zn vacancies may play a role to scatter the carriers and hence lead to a smaller mobility. These two factors result in a poor electrical conductivity for Sr₂ZnSb₂.

Following above discussion, we have deleted this over-interpreted sentence: “The above

two behaviors would undermine the electrical conductivity as demonstrated in Supplementary Fig. 2a,b, similar to the situation of graphite-isostructural *h*-BN (in comparison with the delocalized π bonds in graphite).” In addition, we added a discussion in the caption of Supplementary Fig. 3 to explain the poor electrical conductivity: “It is noted that the Sr₂ZnSb₂ sample has a much lower electrical conductivity in (b). According to the Hall carrier concentration, mobility, and band gap measurements (see Supplementary Table 2), this poor electrical conductivity is attributed to the much lower Hall carrier concentration and much smaller mobility.”

Supplementary Table 2 | Hall carrier concentration, Hall mobility, and electrical band gap of the Sr(Cu,Ag,Zn)Sb compounds at 300 K.

Sample	Hall carrier concentration (cm ⁻³)	Hall mobility (cm ² V ⁻¹ s ⁻¹)	band gap (eV)
SrCuSb	1.2×10 ²⁰	158	0.13
SrAgSb	4.5×10 ¹⁹	254	0.17
Sr ₂ ZnSb ₂	4.3×10 ¹⁸	21	0.12

Responses to Reviewers' comments:

Reviewer #3 (Remarks to the Author):

Zhu et al. investigate the roles of vacancies in thermal transport, using exemplary ZrBeSi-type Zintl phases for their analysis. Zintl phases, particularly those that adopt the ZrBeSi structure type, have recently attracted interest for thermoelectric applications; as a result, elucidating mechanistic roles of lattice dynamics on thermal properties is timely.

However, the reduction in lattice thermal conductivity due to vacancy-induced lattice softening is a well-documented phenomenon, as demonstrated in numerous highly-defective systems (see, e.g., Slade et al., *Joule* 5, 2021). The phenomenon has been cited in the context of thermoelectric performance in ZrBeSi-type phases also (see, e.g., Chanakian et al., *Angew. Chem.* 62, 2023). Moreover, shortcomings in the presented analysis leave room for a more complete analysis of phonon dynamics in ZrBeSi-type Zintl phases.

The paper is interesting, and the neutron diffraction experiments on the Zintl phases are indeed appreciated. Unfortunately, I do not think it provides results of exceptional quality or novelty to merit publication in *Nature Communications*. Please see below for my reasoning.

Response: We thank the Reviewer for the detailed reading and the recognition of the interesting and valuable experimental data. We also appreciate the Reviewer for his/her critical comments and constructive suggestions to improve the rigor in the interpretation of our results. Below are our detailed responses to each of the comments.

We also thank the Reviewer for providing information on the studies of vacancy-induced lattice softening in the work of [Slade *et al.*, *Joule* 5, 1168-1182 (2021)]. We missed this reference in the previous literature review and have added it as a reference in the updated manuscript. As the Reviewer pointed out, the vacancy-induced lattice softening phenomena were indeed reported in the previous researches, such as [Slade *et al.*, *Joule* 5, 1168-1182 (2021)] and [Chanakian, S., *et al. Angew. Chem. Int. Ed.* 62, e202301176 (2023)]. However, these previous works only give brief discussions of the underlying causes for the lattice softening, while the detailed microscopic mechanisms remain elusive.

Nonetheless, in this work, we directly measured the dynamic structural factors in the momentum and energy spaces using the inelastic neutron scattering technique, unveiled the anisotropic atomic vibration through the analysis of atomic displacement parameters based on the neutron powder diffraction patterns, and explored the changes in atomic bonding through charge density analysis and X-ray photoelectron spectroscopy. Based on these spectroscopic techniques and supported by the additional first-principle calculations, we established a novel, in-depth, and comprehensive physical picture for the atomistic mechanism of the vacancy-induced low lattice thermal conductivity in these ZrBeSi-type Zintl phases.

Furthermore, a careful comparison found that the vacancy-induced lattice softening reported in [Slade *et al.*, *Joule* 5, 1168-1182 (2021)] is mainly mediated by charge carriers, and

higher carrier concentrations always lead to lower sound (phonon) velocity, and in some cases, corresponds to the compounds with less vacancy concentration, such as $\text{Nb}_{0.8+x}\text{CoSb}$ and $\text{Pr}_{3-x}\text{Te}_4$. This differs from the phenomena in our work. Here, we found that the vacancy-defective Sr_2ZnSb_2 compound with a softer lattice has a much lower carrier concentration ($\sim 4.3 \times 10^{18} \text{ cm}^{-3}$ at room temperature) than the SrCuSb and SrAgSb counterparts ($1.2 \times 10^{20} \text{ cm}^{-3}$ and $\sim 4.5 \times 10^{19} \text{ cm}^{-3}$). In addition, higher vacancy concentrations are required to get a larger lattice softening. What's more, the softening in Sr_2ZnSb_2 happens to the entire lattice. More importantly, the introduction of vacancies can also induce a strong enhancement of phonon anharmonicity as manifested by the overall phonon broadening and larger Grüneisen parameter. These abnormal but intriguing behaviors result from the changes in the amount of charge transferring between Sr and $[\text{ZnSb}_2]$ sublattices. Therefore, we revealed a different but novel atomistic mechanism for the vacancies to suppress lattice thermal conductivity.

In addition, all the following issues raised by the Reviewer can be properly resolved. The random distribution of vacancies is evidenced by the Rietveld refinements of neutron powder diffraction patterns, the lattice softening could be easily read from the inelastic neutron scattering spectra and the softening of acoustic phonons entails a reduction in the sound (phonon) velocity, and the vacancies is demonstrated to play a defect scattering role at the same time.

Therefore, we believe that the present work represents a novel and significant advance in the fields of thermoelectric ZrBeSi-type Zintl phases and of the vacancy-mediated lattice thermal conductivities, and the revised manuscript is suitable for publication in Nature Communications.

Following above discussion, we made some more discussion in the second paragraph of the “Discussion” Section to distinguish our findings from the previous work in [Slade *et al.*, *Joule* **5**, 1168-1182 (2021)]: “In addition, we would like to point out that the change in the charge transfers causes a sharp drop in the free charge carrier concentration of Sr_2ZnSb_2 ($\sim 4.3 \times 10^{18} \text{ cm}^{-3}$ at room temperature) in comparison to the SrCuSb and SrAgSb counterparts ($1.2 \times 10^{20} \text{ cm}^{-3}$ and $\sim 4.5 \times 10^{19} \text{ cm}^{-3}$). It is reasonable to conclude that this lattice softening in Sr_2ZnSb_2 is also different from the well-studied vacancy-defective SnTe - AgSbTe_2 , $\text{Nb}_{0.8+x}\text{CoSb}$, or $\text{Pr}_{3-x}\text{Te}_4$ compounds, where the softening behavior is essentially associated with the free charge carrier concentration and higher concentration is required to lower the phonon group velocities there.”

Major points:

Comment 1. The absence of long-range ordering in Sr_2ZnSb_2 is stated in the text, but not supported by the provided data. Since this motivated the structural refinement and subsequent simulations (as indicated in lines 106-109), please provide an explanation for the Zn vacancy

ordering.

Response: We thank the Reviewer for this comment and suggestion. In fact, the conclusion of a random distribution of the vacancy in Sr_2ZnSb_2 is obtained from the Rietveld refinement of the neutron powder diffraction (NPD) patterns. In the popular structural analysis with Rietveld refinement, the occupation of an element on a certain Wyckoff position adopt a random distribution in default. If a long-range ordering appears, two situations would happen and the crystal structure model in Rietveld refinement should be altered accordingly. 1) A new crystal symmetry is adopted accompanied with the formation of a superlattice. In this case, extra Bragg peaks would appear in the diffraction patterns. A typical example is the vacancy-ordered double perovskites Cs_2SnI_6 , which adopt a $Fm-3m$ symmetry, different from the simple perovskite structure of $Pm-3m$. 2) A new crystal symmetry is adopted but without the formation of a superlattice. In this case, one cannot observe extra Bragg peaks, but the relative intensities of Bragg peaks would present large differences. One well-known example is the long-range ordering of half vacancy of the 8c Wyckoff position in Heusler alloys with $Fm-3m$. This would cause a structural transformation into a half-Heusler alloys with $F-43m$ symmetry. Although there are not extra Bragg peaks, the relative intensities between different Bragg peaks show giant differences between Heusler and half-Heusler compounds. In this work, it neither requires to consider superlattice nor need a new crystal symmetry to do the structural refinement for Sr_2ZnSb_2 . More importantly, a random vacancy distribution on the Zn site could perfectly fit the NPD pattern, including peak positions and peak intensities. This fact establishes that the vacancies in Sr_2ZnSb_2 are randomly distributed on the Zn site.

In fact, there are more experimental and theoretical results to support this randomly distributed vacancy model:

Experimentally: a random distribution of the vacancies in the similar Eu_2ZnSb_2 is also observed in the previous studies. Besides Rietveld refinement, from the high-angle annular dark-field scanning transmission electron microscopy (HAADF-STEM) image, one can also simultaneously observe locally non-occupied, half-occupied and full-occupied situations [*Sci. Adv.* **7**, eabd6162 (2021)], indicating a random distribution. (Here, it is difficult to run STEM measurement for Sr_2ZnSb_2 as it is much sensitive to air.)

Theoretically: In our previous work [*Chem. Mater.* **34**, 7837-7844 (2022)], it was found that the simulation of lattice thermal conductivity by considering a random vacancy distribution could give a better reproduction of the experimental data (Fig. 6 in the reference). This further confirm the corrections of the randomly distribution. In fact, in this work, we also found that only the randomly distribution model could give a better reproduction to the phonon DOSs measured using INS technique in Fig. 3.

To give a more logical description of this random vacancy distribution in the manuscript, we rewrote the sentences as: “In addition, a random distribution model of the vacancy on the 2d-site could perfectly fit the Sr_2ZnSb_2 NPD patterns, including all the peak positions and peak intensities, as shown in Supplementary Fig. 2c and Fig. 6, indicating an absence of long-range

ordering of the vacancies. This differs from the well-known Cs_2SnI_6 double perovskite where vacancies take a long-range ordering and lead to a change in crystal symmetry and an emergence of superlattice Bragg peaks. In the following context, it will show that the lattice dynamical simulation with random vacancy distribution could give a good reproduction of the experimental phonon density of states (DOSs).”

Comment 2. Regarding the statement on lines 208-209: “Only the Ag-related phonon modes around 10 meV exhibit obviously softening in SrAgSb while the energy for other modes does make large variations.”: the term “softening” in this context does not make sense, as it is a term used to describe the lowering of the phonon group velocity (see, e.g., Hanus et al., Adv. Mater. 31, 2019). The authors even implied that the term “softening” refers to a reduction in the group velocity when comparing the Debye temperatures in line 147. How is the phonon DOS related to the group velocity? If a more general definition of “softening” is adopted in this study, please state that.

Response: Thanks to the Reviewer for pointing out this confusing description. With neutron scattering spectroscopy, a phonon softening is strictly defined as a reduction in the phonon energy, which can be a reduction in the energy of optical phonons or a reduction in the energy of acoustic phonons. It is noted that a decrease in the energy of acoustic phonons represents a decrease in the acoustic phonon group velocity. This can be understood from another perspective: when the lattice symmetry remains unchanged, the phonon DOS of a sample with a slower group velocity of acoustic phonons has a higher intensity in its corresponding energy range, because there are more phonon normal modes accommodated within a unit energy range. In this work, both experimental and simulated acoustic phonon DOSs from 0 to ~3.5 meV show a significant softening or an increase in intensity for the Ag and Zn samples. This demonstrates that the phonon group velocities or sound velocities become smaller in these two samples. (The acoustic phonon branches could extend up to ~10 meV, and they overlap with low-energy optical phonon. Therefore, when discuss the softening behavior of group velocities of the acoustic phonons, we should focus on the 0 to ~3.5 meV, where the phonon DOSs only contain the acoustic phonons).

Following this discussion, at the end of the first paragraph of the section “Lattice dynamics and strong phonon anharmonicity”, we added an important comment: “In addition, it is noted that the phonon DOSs from 0 to ~3.5 meV also show obvious softening in energy or large increase in intensity, demonstrating a lowering of the acoustic phonon group velocities and agreeing with the experimental results of sound velocities and the Debye temperatures in Table 1.”

The theoretical calculation results in the third paragraph of the section “Lattice dynamics and strong phonon anharmonicity” were employed to rationalize the inelastic neutron scattering experimental data. It mainly emphasized that the phonon softenings in SrAgSb and Sr_2ZnSb_2 are essentially different. Nonetheless, in SrAgSb, the Ag-related phonon modes from ~3 to ~10

meV contain the top part of the acoustic phonons and the low-lying optical phonons. Their softening also clearly implies the lowering of the phonon group velocity. We modified this sentence as: “In comparison to SrCuSb, only the Ag-related phonon modes below ~10 meV exhibit obviously softening in SrAgSb while the energy for other modes does not make large variations.”

Comment 3. The reduction in lattice thermal conductivity in Sr₂ZnSb₂, compared to SrCuSb and SrAgSb, is mainly attributed to lattice softening induced by vacancies in this study. However, the role of defect scattering, which can significantly affect the lattice thermal conductivity as shown in papers cited by the authors (e.g., Refs 29 and 30), is not fully investigated in this study. Further analysis should be provided to thoroughly elucidate the mechanisms underlying the low lattice thermal conductivity in the 2-1-2 phase.

Response: We thank the Reviewer for this interesting and important comment. We also thank the Reviewer for encouraging us to learn more about the analysis of the phonon-vacancy scattering in Refs 29 and 30.

In these two references, the influence of the vacancies on lattice thermal conductivity is analyzed using a Callaway-type model by considering the phonon-phonon, phonon-grain boundary, phonon-point defect, and phonon-vacancy scatterings, as well as the group velocity softening. A careful read would find that the phonon-vacancy scattering is assumed to be different from the phonon-point defect scattering. This assumption comes from the discussion by Ratsifaritana and Klemens in [*Int. J. Thermophys.* **8**, 737-750 (1987)]. They demonstrated that the vacancy defect does not bring strong strain-fluctuation scattering and low-lying resonance scattering. This is different from other kinds of point defects because the vacancies would break the linkage from the surrounding atoms. Therefore, it can use the perturbation theory to properly deal with the mass-fluctuation scattering by assuming $\Delta M = 3M$ (M is the average atomic mass). The successful analysis of the SnTe-AgSbTe₂ and SnTe-NaSbTe₂ has two prerequisites. One is that the vacancy concentration is much small so the perturbation theory is still valid. For example, AgSn_mSbTe_{2+m} compounds have a vacancy concentration from 0.7% to 4.6% upon alloying of AgSbTe₂ ($m = 5$). The other presupposition is that the vacancy concentration could be tuned in SnTe but without introducing other elements and point defect scattering, and this could be used to accurately determine the phonon-vacancy scattering strength of s^2 .

In the case of Sr₂ZnSb₂, one main issue for a comprehensive analysis of the roles of vacancy on the suppression of lattice thermal conductivity is to determine whether it is a phonon-vacancy scattering or a general phonon-point defect scattering. As suggested by the reference of [*Int. J. Thermophys.* **8**, 737-750 (1987)], we should determine if the Sr₂ZnSb₂ shows an internal strain field. Therefore, we compared the full width at half maximum (FWHM) of the Bragg peaks in the neutron powder diffraction patterns. As shown in Response Fig. 3, the (110) Bragg peak of the Sr₂ZnSb₂ compound has a much larger FWHM, demonstrating that

the vacancies in Sr_2ZnSb_2 also present a strong strain field. This behavior confirms that the vacancies in Sr_2ZnSb_2 play a phonon-point defect scattering role rather than a phonon-vacancy scattering role as in SnTe [*Int. J. Thermophys.* **8**, 737-750 (1987); *ACS Energy Lett.* **3**, 705-712 (2018); *J. Am. Chem. Soc.* **142**, 12524-12535 (2020)].

Response Fig. 3 | The comparison of the example Bragg peak of (110) for the $\text{Sr}(\text{Cu,Ag,Zn})\text{Sb}$ compounds. The data were collected at 300 K on the SuperHRPD high resolution neutron diffractometer at J-PARC in Japan.

However, a quantitative analysis to separate the contribution to the suppression of lattice thermal conductivity from the softening phonon group velocity, the phonon-phonon anharmonicity scattering, and the phonon-point defect scattering is a challenging issue in the current case. One difficulty is to build a proper and accurate physical model to describe the vacancy-induced strain field. A side question following this is that whether the perturbation theory is still valid in deal with the point scattering in Sr_2ZnSb_2 as the vacancy concentration is too high. Another one is that both phonon anharmonicity and point defect scatterings are introduced in Sr_2ZnSb_2 in comparison to the SrCuSb , and a simultaneous determination of these two variables is difficult. A proper reference sample, different vacancy concentration samples, and more experimental characterization are required. This is far beyond the main scope of this work, although a further and comprehensive study is worthwhile.

Nonetheless, to better understand the scattering mechanisms, we still try to use the Debye-Callaway model to analyze the lattice thermal conductivity with phonon relaxation time contributed by different scattering processes:

$$\kappa_L = \int_0^{\omega_D} \kappa_S(\omega) d\omega = \frac{k_B}{2\pi^2 v} \left(\frac{k_B T}{\hbar} \right)^3 \int_0^{\theta_D/T} \tau \frac{x^4 e^x}{(e^x - 1)^2} dx \quad (\text{R8})$$

where k_B , v , \hbar , T , θ_D and are the Boltzmann constant, average sound velocity sound velocity, Planck constant, absolute temperature, Debye temperature, and phonon relaxation time, respectively. x is the reduced phonon frequency $x = \hbar\omega/k_B T$. The phonon relaxation time τ can be obtained by considering the contributions from different scattering processes:

$$\tau^{-1} = \tau_U^{-1} + \tau_N^{-1} + \tau_B^{-1} + \tau_V^{-1} \quad (\text{R8})$$

where τ_U , τ_N , τ_B and τ_V refer to the phonon relaxation time for Umklapp scattering(U), normal scattering (N), boundary scattering(B), and vacancy-related scattering (V), respectively. In this analysis, we set SrAgSb as pristine material. The umklapp scattering, normal scattering, and boundary scattering are considered in SrAgSb. For the Sr₂ZnSb₂, the umklapp scattering, normal scattering, boundary scattering, and vacancy-related scattering are considered. As shown in Response Fig. 4, both lattice softening and vacancy scattering contribute to the low lattice thermal conductivity of Sr₂ZnSb₂. Here, the term of ‘vacancy-related scattering’ is used to distinguish from the phonon-vacancy scattering as introduced in SnTe [*ACS Energy Lett.* **3**, 705-712 (2018); *J. Am. Chem. Soc.* **142**, 12524-12535 (2020)].

Response Fig. 4 | Lattice thermal conductivity modeling using Debye-Callaway model. The SrAgSb is taken as the pristine materials, and it is clear that the vacancy-related scatterings make large contribution to the suppression of the lattice thermal conductivity in the vacancy-defective Sr₂ZnSb₂.

In short, we can judge that the vacancy also plays a point defect scattering rather than a vacancy scattering as in SnTe. However, an accuracy determination of the contribution from different physical mechanisms may need further investigation. Following this discussion, we modified last sentence in the first paragraph in the Discussion section as: “In short, the strongly enhanced phonon softening and anharmonicity, together with **phonon-point defect scatterings**, endue the Sr₂ZnSb₂ compound with a much smaller lattice thermal conductivity.”

Minor points:

Comment 4. 1. Regarding the statement on lines 156-157: “It is easy to read from Fig. 2g that the optical phonon modes in both Ag and Cu compounds shift towards lower energy.”: it is unclear how it can be understood that optical phonon modes have lower frequencies in SrCuSb and SrAgSb. Please provide a short explanation.

Response: Firstly, there is a typo error here, and it is the optical phonon modes in the Ag and

Zn compounds that shift towards lower energy.

Secondly: the acoustic phonon contribution to specific heat in the Debye model is proportional to T^3 , and the C_p/T^3 vs T curve should be constant at low temperatures where only the acoustic phonons are thermally excited. However, at higher temperature, following the appearance of the thermally excited optical phonons, C_p does no longer follow the Debye T^3 law. In this situation, one need to take the Einstein model into account and a Boson-like hump may appear. The hump peak position represents the temperature threshold to thermally excite the optical phonon. Therefore, lower peak position of the hump means lower frequency of the optical phonon modes or lower Einstein temperatures.

Following this response, we made revision to the paragraph: “For polyatomic compounds, the contribution of optical phonons to the thermal properties should be included. This necessity is demonstrated through the appearance of a Boson-like hump in the C_p/T^3 vs T plot in Fig. 2g, since this hump feature implies a deviation of C_p from the Debye T^3 law. Furthermore, the hump peak position in the C_p/T^3 vs T plot corresponds to a crossover of the contribution to heat capacity from the acoustic phonons to optical phonons, and the lower peak positions in Fig. 2g indicate lower energies of the optical phonons in both the Ag and Zn compounds than in the Cu compound. The θ_D and θ_E (Einstein temperatures) from the fit of the B_{ov} and the C_p data in Fig. 2 are listed in Table 1. A careful comparison of the three Einstein temperatures...”

Comment 5. Why is the PBE functional used for SrCuSb and SrAgSb, but PBEsol for Sr₂ZnSb₂? Would the usage of different DFT functionals lead to inconsistent results? Please provide an explanation.

Response: We chose the PBEsol exchange-correlation functional for Sr₂ZnSb₂ based on the consistency with previous literature [*Chem. Mater.* **34**, 7837-7844 (2022)] and the DFT-optimized lattice parameters. In the previous investigation, the DFT-equilibrated lattice parameters for the primitive cell of Sr₂ZnSb₂ with different exchange-correlational functionals have been calculated and listed as follows. A more reasonable agreement has been found between theoretically predicted lattices with the PBEsol exchange-correlational functional and the X-ray diffraction (XRD) and neutron powder diffraction (NPD) data.

Response Table 2 | The lattice parameters obtained from simulations using LDA, PEB, and PBEsol, as well as the data extracted from the analysis of XRD and NPD patterns.

Methods	a (Å)	c (Å)
LDA	4.545	8.393
PBE	4.695	8.821
PBEsol	4.605	8.540
XRD (room temperature) [Anorg. Allg. Chem. 637 , 2018–2025 (2011)]	4.6234(13)	8.345(3)
NPD (300 K) [this work]	4.6645(3)	8.3833(14)

Following this comment, we added an explanation in the Methods: “The Perdew-Burke-Ernzerhof (PBE) of generalized gradient approximation (GGA-PBE) was applied as exchange-correlation functional for SrCuSb and SrAgSb, while PBEsol was adopted for Sr₂ZnSb₂ which could more reasonably reproduce the lattice parameters extracted from the experimental data.”

Comment 6. Are the phonon DOS in Fig. 3e calculated using two different methods? If so, is this because the random distribution of Zn vacancies in Sr₂ZnSb₂ breaks periodicity? Please explain why different methods are used.

Response: Yes, the main reason is related to the breakdown of periodicity due to the random distribution of Zn vacancies. The TDEP approach leverages lattice symmetry to mitigate computational demands and numerical errors during the renormalization process. However, its accuracy and effectiveness diminish while dealing with the SQS structure with disordered vacancies due to the loss of symmetry. Therefore, we calculated the power spectra from the autocorrelation function of velocities, which allows us to account for full anharmonicity and disorder.

Following this comment, we added an explanation in the Methods: “The TDEP approach exploits the symmetry of the lattice to reduce computational workload and improve the numerical accuracy during the renormalization process. This approach becomes genuinely disadvantageous in dealing with the Zn-compound, where the randomly distributed vacancies break the symmetry. On the other hand, the autocorrelation function of velocities based on the equilibrium molecular dynamics (EMD) simulations allows one to simultaneously account for full anharmonicity and occupancy disorder. Therefore, while the lattice dynamics of SrCuSb and SrAgSb were simulated with TDEP, the moment tensor potential (MTP) of Sr₂ZnSb₂ were built from EMD simulations under the NVE ensemble using the LAMMPS package at 100 and 300 K.”

Comment 7. Do the Grüneisen parameters for the three compounds listed in Table 1 match with DFT-calculated values? Such an analysis can strengthen the argument that Sr₂ZnSb₂ exhibits stronger anharmonicity than SrCuSb and SrAgSb, as well as provide further credence towards the computational results as a whole.

Response: We thank the Reviewer for this suggestion. We have carefully reviewed the origin of the physical model of the sound velocity method used to estimate Grüneisen parameters (see Supplementary Equation 4). This method presumes that the phonon DOS is simplified as a Debye model. This might bring large inaccuracy in estimating the Grüneisen parameters for SrAgSb and Sr₂ZnSb₂ as these two samples have much lower optical phonons that overlap with the acoustic phonon. Therefore, to make the comparison of Grüneisen parameters between the three samples more convincing, we also estimate the Grüneisen parameters in this work from the phonon density of states (DOS), $g(E)$, as determined from inelastic neutron scattering measurement. An average Grüneisen parameter, $\bar{\gamma}$, could be obtained with the following

formula [*Phys. Rev. B* **80**, 184302 (2009)] of $\bar{\gamma} = d \ln \langle E \rangle / \ln V$, where $\langle E \rangle = \int E g(E) dE$ is the average phonon energy and V is unit cell volume. Combined with the unit cell volume obtained from the refinement of NPD data, $\bar{\gamma}$ of SrCuSb, SrAgSb, and Sr₂ZnSb₂ are estimated as 0.769, 1.739, and 2.799, respectively. The Sr₂ZnSb₂ shows a larger value than Sr(Cu,Ag)Sb, corroborating the conclusions of this work that vacancies bring about stronger phonon anharmonicity in Sr₂ZnSb₂. This enlarged phonon anharmonicity also manifests as an overall broadening of the phonon DOSs in Fig. 3 of the main text. Furthermore, the larger three-phonon and four-phonon scattering rates of Sr₂ZnSb₂ further corroborate this finding (refer the discussion in Supplementary Fig. 16).

To properly respond to this comment, we also calculated the Grüneisen parameters for SrCuSb and SrAgSb at 0 K using quasi-harmonic approximation with a volume change of ± 1 %. However, we cannot take a similar calculation for Sr₂ZnSb₂ within such a finite difference method, as its accuracy diminishes while dealing with the SQS structure with disordered vacancies due to the loss of symmetry. Nonetheless, the theoretical value of the Grüneisen parameter for SrAgSb in Response Fig. 5 shows an overall larger trend than the SrCuSb. This agrees with the trends of the average Grüneisen parameters as estimated from experimental phonon DOSs, confirming the validity of the conclusion built on the average Grüneisen parameters.

Response Fig. 5 | Grüneisen parameters of SrCuSb and SrAgSb at 0 K calculated using quasi-harmonic approximation with a volume change of ± 1 %.

Following this response, the $\bar{\gamma}$ values are summarized in Table 1 in the revised manuscript. In addition, we modified the discussion in the second paragraph of the “Lattice dynamics and strong phonon anharmonicity” section as: “Moreover, the sound velocity analysis yields close Grüneisen parameters for SrCuSb (1.42) and SrAgSb (1.43), but the Sr₂ZnSb₂ gives a larger value of 1.50 (Table 1). This enlarged behavior of the phonon Grüneisen parameter in Sr₂ZnSb₂ is also captured by the average Grüneisen parameters, $\bar{\gamma} = d \ln \langle E \rangle / \ln V$, where $\langle E \rangle = \int E g(E) dE$ is the average phonon energy (Fig. 3f) and V is unit cell volume determined from NPD patterns. Here, the different absolute values of the Grüneisen parameter from these two methods might come from the over-simplified Debye model used in the sound velocity method.”

Spelling errors (which did not affect the decision):

1. Line 106: “Wort” is misspelled.

Response: Thank the Reviewer for figuring out this misspelling. We corrected it in the revised manuscript: “It is **worth** noting that no long-range ordering or superlattice are observed for...”

2. Line 409: “Ernzerh” is misspelled.

Response: The typo has been corrected: “The **Perdew-Burke-Ernzerhof** of generalized gradient approximation (GGA-PBE) was applied ...”

REVIEWER COMMENTS

Reviewer #1 (Remarks to the Author):

I thank the authors for responding to my comments in detail. I am satisfied with them for the most part.

But I think the authors likely misinterpreted my comment#3, for which I apologize if I wasn't being clear. The corrected statement from the author reads "All of the above factors are intrinsic properties and mainly depend on a given material's crystal symmetry and chemical components. However, it is difficult to modify them artificially within the valid range of perturbation theory." But no, it is entirely possible to modify them to within the regime of perturbation theory - just be very dilute. What I meant is that, any changes made to the lattice to a sufficiently large extent will alter its intrinsic behaviors. Therefore the authors should remove the statement that intrinsic properties can't be modified. It is only when the change is so dilute that, in the theoretical jargon, perturbation theory can be applied that one can "assume" that intrinsic properties doesn't change.

Reviewer #2 (Remarks to the Author):

The authors have diligently addressed all of my concerns. Consequently, I recommend the manuscript for publication.

Reviewer #3 (Remarks to the Author):

The authors clarified most of my concerns. I agree with the author's response that the niche aspect of the work is to provide an atomistic mechanism of the lattice thermal conductivity reduction in these Zintl phases, and the results certainly contribute useful insight for the most part.

The analysis of phonon-defect scattering in Sr₂ZnSb₂ (discussed in response to Comment #3) provides a more holistic understanding, particularly Response Fig. 4 which quantitatively shows the relative contributions from lattice softening and phonon-defect scattering to the lowering of lattice thermal conductivity. In my opinion, this analysis provides fascinating insight and deserves further discussion in the text. Excluding this analysis from the paper would be somewhat contradictory to the overall goal of providing a comprehensive, mechanistic explanation for the reduction in lattice thermal conductivity.

Before recommending publication to Nature Communications, please justify the omission.

Responses to Reviewers' comments:

Reviewer #1 (Remarks to the Author):

I thank the authors for responding to my comments in detail. I am satisfied with them for the most part.

Response: We thank the Reviewer for the careful review of our work again, and for this high evaluation on our revised manuscript.

But I think the authors likely misinterpreted my comment#3, for which I apologize if I wasn't being clear. The corrected statement from the author reads "All of the above factors are intrinsic properties and mainly depend on a given material's crystal symmetry and chemical components. However, it is difficult to modify them artificially within the valid range of perturbation theory." But no, it is entirely possible to modify them to within the regime of perturbation theory - just be very dilute. What I meant is that, any changes made to the lattice to a sufficiently large extent will alter its intrinsic behaviors. Therefore the authors should remove the statement that intrinsic properties can't be modified. It is only when the change is so dilute that, in the theoretical jargon, perturbation theory can be applied that one can "assume" that intrinsic properties doesn't changed.

Response: We thank Reviewer for this detailed explanation. Following this suggestion, we have deleted the improper statement that intrinsic properties can't be modified: "All of the above factors are intrinsic properties and mainly depend on a given material's crystal symmetry and chemical components. On the other hand, importing extrinsic defects is a controllable strategy to....."

Reviewer #2 (Remarks to the Author):

The authors have diligently addressed all of my concerns. Consequently, I recommend the manuscript for publication.

Response: We thank the Reviewer for his/her review of our work for the second time, and for the positive comment on our revised manuscript.

Reviewer #3 (Remarks to the Author):

The authors clarified most of my concerns. I agree with the author's response that the niche aspect of the work is to provide an atomistic mechanism of the lattice thermal conductivity reduction in these Zintl phases, and the results certainly contribute useful insight for the most part.

Response: We thank the Reviewer for the constructive comments on our work, and for the positive evaluation on our revised manuscript.

The analysis of phonon-defect scattering in Sr₂ZnSb₂ (discussed in response to Comment #3) provides a more holistic understanding, particularly Response Fig. 4 which quantitatively shows the relative contributions from lattice softening and phonon-defect scattering to the lowering of lattice thermal conductivity. In my opinion, this analysis provides fascinating insight and deserves further discussion in the text. Excluding this analysis from the paper would be somewhat contradictory to the overall goal of providing a comprehensive, mechanistic explanation for the reduction in lattice thermal conductivity.

Before recommending publication to Nature Communications, please justify the omission.

Response: We thank the Reviewer for this comment. Following this suggestion, we added one paragraph in the Discussion section to provide a holistic explanation on the reduction in lattice thermal conductivity of the vacancy-defective Sr₂ZnSb₂: “In addition, the vacancies in the SnTe-AgSbTe₂ compounds were considered to play a phonon-vacancy scattering role at the same time. Different from traditional point defect scattering, the vacancy scattering only considers the mass-fluctuation scattering but ignores strain-fluctuation scattering. However, our high-resolution NPD patterns indicate that Sr₂ZnSb₂ has a much stronger strain field in as it exhibits a larger Bragg peak width than Sr(Cu,Ag)Sb (Supplementary Fig. 17). Therefore, the vacancies in Sr₂ZnSb₂ should also play a role in phonon-point defect scattering in addition to softening the phonon group velocities. To better understand the scattering mechanisms, we analyze the lattice thermal conductivity SrAgSb and Sr₂ZnSb₂ using the Debye-Callaway model (Supplementary Note 4). When we only considered the phonon scattering, as shown in Supplementary Fig. 18, the simulated lattice thermal conductivity of Sr₂ZnSb₂ is lower than that of SrAgSb, indicating that lattice softening contributes to the low lattice thermal conductivity of Sr₂ZnSb₂. In addition, vacancy-related scattering is also important to achieve ultralow lattice thermal conductivity with a weak temperature dependence for Sr₂ZnSb₂.”

In addition, more details behinds the discussion are given in Supplementary Note 4, Supplementary Figs. 17&18:

“Supplementary Note 4 | Debye-Callaway model analyses

Debye-Callaway model was used to analyze the lattice thermal conductivity with phonon relaxation time contributed by different scattering processes:

$$\kappa_{\text{lat}} = \int_0^{\omega_D} \kappa_S(\omega) d\omega = \frac{k_B}{2\pi^2 v} \left(\frac{k_B T}{\hbar} \right)^3 \int_0^{\theta_D/T} \tau \frac{x^4 e^x}{(e^x - 1)^2} dx \quad (6)$$

where k_B , v , \hbar , T , θ_D and τ are the Boltzmann constant, average sound velocity, Planck constant, absolute temperature, Debye temperature, and phonon relaxation time, respectively. x is the

reduced phonon frequency $x = \hbar\omega/k_B T$. The phonon relaxation time τ can be obtained by considering the contributions from different scattering processes:

$$\tau^{-1} = \tau_U^{-1} + \tau_N^{-1} + \tau_B^{-1} + \tau_V^{-1} \quad (7)$$

where τ_U , τ_N , τ_B and τ_V refer to the phonon relaxation time for Umklapp scattering (U), normal scattering (N), boundary scattering (B), and vacancy-related scattering (V), respectively. In this analysis, we set SrAgSb as pristine material. The Umklapp scattering, normal scattering, and boundary scattering are considered in SrAgSb. For the Sr₂ZnSb₂, the Umklapp scattering, normal scattering, boundary scattering, and vacancy-related scattering are considered. The results are shown in Supplementary Fig. 18.

Supplementary Fig. 17 | The comparison of the example Bragg peak of (110) for the Sr(Cu,Ag,Zn)Sb compounds. The data were collected at 300 K on the SuperHRPD high resolution neutron diffractometer at J-PARC in Japan. As shown in this, the example (110) Bragg peak of the Sr₂ZnSb₂ compound has a much larger full width at half maximum (FWHM), demonstrating that the vacancies in Sr₂ZnSb₂ also present a strong strain field. This behavior confirms that the vacancies in Sr₂ZnSb₂ play a phonon-point defect scattering role rather than a phonon-vacancy scattering role as in SnTe. In SeTe, phonon-vacancy scattering physical picture considers that the vacancy only induces a mass-fluctuations in the matrix but without strain fluctuation, while point defect scattering contain both fluctuations.

Supplementary Fig. 18 | Lattice thermal conductivity modeling using Debye-Callaway model. The SrAgSb is taken as the pristine materials, and it is clear that the vacancy-related scatterings make large contribution to the suppression of the lattice thermal conductivity in the vacancy-defective Sr₂ZnSb₂. More details about the fitting are given in Supplementary Note 4.”

REVIEWERS' COMMENTS

Reviewer #1 (Remarks to the Author):

The authors have addressed my comments and I now recommend publication without further changes.

Reviewer #3 (Remarks to the Author):

The authors addressed my concern and sufficiently revised the manuscript. A more comprehensive discussion of the phonon behavior is now provided. I recommend the manuscript in its current form.

Responses to Reviewers' comments:

Reviewer #1 (Remarks to the Author):

The authors have addressed my comments and I now recommend publication without further changes.

Response: We thank the Reviewer again for the helpful comments.

Reviewer #3 (Remarks to the Author):

The authors addressed my concern and sufficiently revised the manuscript. A more comprehensive discussion of the phonon behavior is now provided. I recommend the manuscript in its current form.

Response: We are grateful to the Reviewer for the careful review of our manuscript and for the suggestive comments to improve our work.